# Communication between the stem cell niche and an adjacent differentiation niche through miRNA and EGFR signaling orchestrates exit from the stem cell state in the *Drosophila* ovary

**Jiani Chen**[1,2,3], **Chaosqun Li**[1,2], **Yifeng Sheng**[1,2], **Junwei Zhang**[1,2], **Lan Pang**[1,2,3], **Zhi Dong**[1,2], **Zhiwei Wu**[1,2], **Yueqi Lu**[1,2], **Zhiguo Liu**[1,2], **Qichao Zhang**[1,2,3], **Xueying Guan**[3,4], **Xuexin Chen**[1,2,4], **Jianhua Huang**[1,2]*

1 Institute of Insect Sciences, Ministry of Agriculture Key Lab of Molecular Biology of Crop Pathogens and Insect Pests, Zhejiang University, Hangzhou, China, 2 Key Laboratory of Biology of Crop Pathogens and Insects of Zhejiang Province, Zhejiang University, Hangzhou, China, 3 Zhejiang Provincial Key Laboratory of Crop Genetic Resources, Institute of Crop Science, Plant Precision Breeding Academy, College of Agriculture and Biotechnology, Zhejiang University, Hangzhou, China, 4 Hainan Institute of Zhejiang University, Yazhou Bay Science and Technology City, Sanya, China

* jhhuang@zju.edu.cn

**Data Availability Statement:** The transcriptome data have been deposited in the National Center for Biotechnology Information with accession no.

## Abstract

The signaling environment, or niche, often governs the initial difference in behavior of an adult stem cell and a derivative that initiates a path towards differentiation. The transition between an instructive stem cell niche and differentiation niche must generally have single-cell resolution, suggesting that multiple mechanisms might be necessary to sharpen the transition. Here, we examined the *Drosophila* ovary and found that Cap cells, which are key constituents of the germline stem cell (GSC) niche, express a conserved microRNA (miR-124). Surprisingly, loss of miR-124 activity in Cap cells leads to a defect in differentiation of GSC derivatives. We present evidence that the direct functional target of miR-124 in Cap cells is the epidermal growth factor receptor (EGFR) and that failure to limit EGFR expression leads to the ectopic expression of a key anti-differentiation BMP signal in neighboring somatic escort cells (ECs), which constitute a differentiation niche. We further found that Notch signaling connects EFGR activity in Cap cells to BMP expression in ECs. We deduce that the stem cell niche communicates with the differentiation niche through a mechanism that begins with the selective expression of a specific microRNA and culminates in the suppression of the major anti-differentiation signal in neighboring cells, with the functionally important overall role of sharpening the spatial distinction between self-renewal and differentiation environments.

## Introduction

All adult stem cells have the potential to divide and to produce differentiated derivatives [1,2]. Differentiation typically initiates immediately after stem cell division in paradigms with

PRJNA859686. All other data in this study are included in the article and/or supporting information.

**Funding:** This work was supported by the National Natural Science Foundation of China 32325044 and 32172467 to J.H., the China Postdoctoral Science Foundation 2023T160573 and 2022M710127 to J. C., and the National Natural Science Foundation of China U22A20485 to X.C. The funders had no role in study design, data collection and analysis, decision to publish, or preparation of the manuscript.

**Competing interests:** The authors have declared that no competing interests exist.

**Abbreviations:** BMP, bone morphogenetic protein; CB, cystoblast; Dpp, decapentaplegic; EC, escort cell; EGFR, epidermal growth factor receptor; FACS, fluorescence-activated cell sorting; FSC, follicle stem cell; GSC, germline stem cell; UTR, untranslated region.

uniformly long-lived stem cells ("single-cell asymmetry") or independent of division in paradigms where stem cells have variable longevity and exhibit stochastic behaviors ("population asymmetry") [3–6]. In a few cases of single-cell asymmetry, as in neuroblasts, the outcome is determined by asymmetric inheritance from the parent stem cells [7]. More commonly, differentiation depends on a change in cell location and the associated signaling environment. Initial differences in position and signaling may be small. It might therefore be expected that there are mechanisms in place that facilitate the translation of these differences into robust decisions despite inherent biological variability. Conversely, changing environmental conditions or stresses can in some cases provoke purposeful de-differentiation of cells back to a stem cell state, and this might be facilitated if immediate derivatives remain poised to respond to stem cell niche signals. Germline stem cells (GSCs) in the *Drosophila* ovary provide an exceptional paradigm for studying the precise interplay, over short distances and times, of signals controlling the balance between stem cell maintenance and differentiation [8–10].

There are about 15 independent developmental units, termed ovarioles in each *Drosophila* ovary. The most anterior region of each, called the germarium, houses 2 or 3 GSCs and multiple somatic stem cells, known as follicle stem cells (FSCs). These 2 stem cell populations experience overlapping sets of extracellular signals and must coordinate their activities, but both their organization and responses to specific signals are quite different [11–15]. Each GSC most commonly divides with asymmetric outcomes to produce another GSC and a cystoblast (CB) that continues on to further differentiation without delay. CB differentiation involves 4 synchronized divisions with incomplete cytokinesis to form 2-, 4-, 8-, and 16-cell cysts (Fig 1A). Sixteen-cell cysts are subsequently surrounded by derivatives of FSCs and then bud from the posterior of the germarium to form an egg chamber that includes a monolayer follicle cell epithelium. By contrast, FSCs are maintained by population asymmetry [16].

During division, each GSC remains in position at the anterior of the germarium in direct contact with non-dividing somatic Cap cells, while the prospective "daughter CB" is projected posteriorly away from Cap cells. The different behavior of the 2 products of GSC division is largely instructed by their different environments, rather than by asymmetric inheritance (Fig 1A) [17,18]. Specifically, Cap cells produce a Decapentaplegic (Dpp) signal, which is very restricted in its range and activates a bone morphogenetic protein (BMP) signaling cascade in adjacent GSCs to repress the transcription of a key differentiation gene, *bag of marbles* (*bam*), thereby maintaining GSC identity [19–21].

Differentiating CBs and their derivatives are enveloped by somatic, non-dividing escort cells (ECs). Most ECs produce little or no Dpp. ECs also have only limited production of factors that would stabilize or promote the spread of Dpp made in Cap cells. Consequently, BMP signaling is much weaker in CBs and their derivatives than in GSCs, permitting Bam expression and differentiation [20,22]. The properties of ECs that limit BMP signaling in neighboring germline cells are not merely a default state; they are actively enforced by several identified chromatin modifiers and signaling pathways, as well as requiring the maintenance of EC processes, which wrap around germline cells to ensure intimate contact. ECs have therefore been termed a differentiation niche. All differentiation actions of ECs determined to date involve restriction of BMP signaling [23–27].

The most anterior ECs are exceptional. They are located in a transition zone and have been found occasionally to express significant levels of Dpp, supplementing Cap cell signals to GSCs but insufficient to prevent differentiation of CBs [28,29]. Under conditions conducive to de-differentiation, Dpp production from anterior ECs facilitates CB reversion to GSCs, perhaps explaining why low level Dpp production is advantageous [29]. Thus, the key Dpp niche signal has a carefully orchestrated, limited range of signaling to allow rapid onset of differentiation of non-stem cell daughters, while a limited extension of signaling beyond GSCs facilitates the ability to reverse course under altered conditions.

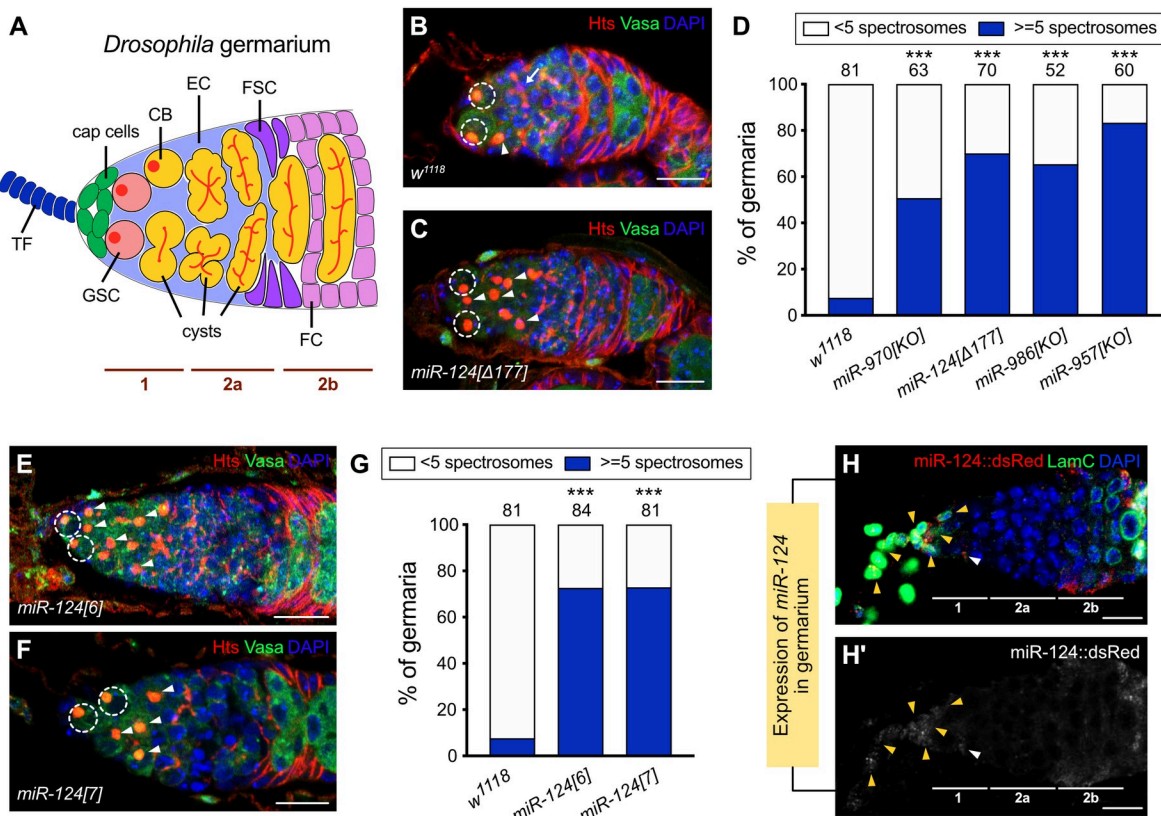

**Fig 1. miR-124 affects GSC daughter cell differentiation. (A)** Illustration of the cellular composition of the *Drosophila* germarium. The cell types are TF (dark blue); Cap cell (green); GSC (light red); CB (orange); cyst (orange); EC (light blue); FSC (purple), and FC (light purple). Both GSC and its CB daughter have a spectrosome (red round dot), whereas cysts have a fusome (red branched structure) instead. **(B and C)** Germaria for *w^1118* wild-type **(B)** or *miR-124[Δ177]* mutant **(C)** immunostained for Hts (red, spectrosomes and fusomes), Vasa (green, germline cells), and DAPI (blue, nuclei). Scale bar: 10 μm. The control germarium **(B)** contained 3 spectrosome-containing cells, including 2 GSCs (indicated by dashed circle) and 1 CB (indicated by arrowhead). The fusome of the differentiating cyst is indicated by an arrow. The germarium of the *miR-124[Δ177]* mutant **(C)** contained extra spectrosome-containing cells. The GSCs are highlighted by dashed circles, whereas the CBs or CB-like single germ cells are indicated by arrowheads. **(D)** Percentage of the germaria carrying 5 or more spectrosome-containing cells for wild-type (*w^1118*) and 4 miRNA mutants showing the GSC daughter differentiation defect. The number of analyzed germaria is shown above each bar. Significance was determined by Fisher's exact two-sided test (*** *P* < 0.001). **(E and F)** Germaria of 2 independent *miR-124* mutant lines, *miR-124[6]* **(E)** and *miR-124[7]* **(F)**, contained excess spectrosome-containing cells and were immunostained for Hts (red), Vasa (green), and DAPI (blue). In **(E)** and **(F)**, the GSCs are highlighted by dashed circles, whereas the CBs or CB-like cells are indicated by arrowheads. Scale bar: 10 μm. **(G)** Percentage of the germaria carrying 5 or more spectrosome-containing cells of wild-type (*w^1118*) and 2 independent miR-124 null allele lines *miR-124[6]* and *miR-124[7]*. The number of analyzed germaria is shown above each bar. Significance was determined by Fisher's exact two-sided test (*** *P* < 0.001). **(H and H')** Expression of miR-124 indicated by a *miR-124*:DsRed (red) reporter in germaria, which was stained with the TF and Cap cell markers LamC (green) and DAPI (blue). **(H')** is the black and white version of **(H)**. MiR-124 is expressed in only TF and Cap cells (yellow arrowhead) but shows some expression in the most anterior ECs (white arrowhead). Scale bar: 10 μm. The raw data underlying panels D and G are available in S1 Data. CB, cystoblast; EC, escort cell; FC, follicle cell; FSC, follicle stem cell; GSC, germline stem cell; TF, terminal filament.

We wished to explore further how a stem cell niche and an adjacent differentiation niche collaborate to orchestrate exit from the stem cell state. Specifically, we focussed attention on microRNAs as potential mediators of fine-scale regulation because of their potential to modulate or stabilize biological states through the posttranscriptional regulation of gene expression [30,31]. MicroRNAs (miRNAs) are 21–25 nucleotide (nt) cellular RNAs that negatively regulate gene expression at the posttranscriptional level by binding to the 3′-untranslated region (UTR) of cognate mRNAs [32–34]. Primary miRNA (pri-miRNA) transcripts are transformed into mature miRNA by the successive actions of 2 RNase III endonucleases. Drosha converts

pri-miRNA transcripts into precursor miRNA (pre-miRNA); Dicer-1, in turn, converts pre-miRNA into mature miRNA [35,36].

MicroRNAs have already been shown to play an important role in GSCs. The disruption of *Dicer-1* and *Drosha* in GSCs leads to a significant reduction in cell division and to the disruption of GSC maintenance [37–39]. Other components involved in miRNA biogenesis and functions, such as loquacious (*loqs*) or Argonaute-1 (*Ago1*), are also required for GSC maintenance [40–42]. Further insights have been gained by studying specific microRNAs. One example is bantam, without which GSCs are rapidly lost [43]. Conversely, the absence of miR-184 in GSC lineages increases the levels of the Dpp receptor Saxophone (*Sax*), leading to *bam* repression in GSC progeny and delayed differentiation [44].

Here, we investigated miRNA functions in GSC niche cells, starting with a genetic screen and then continuing to mechanistic studies for a specific conserved miRNA identified as important for GSC differentiation. We found that the critical function of miR-124 is to limit *dpp* transcription in ECs, by reducing EGFR expression in Cap cells. Further tests showed that Notch signaling was necessary for Cap cells with absent miR-124 and elevated EGFR to elicit ectopic *dpp* expression in ECs. The discovery of regulated interactions between adjacent stem cell and differentiation niches reveals a previously unexplored tier of control over stem cell differentiation.

## Results

### A miRNA screen identifies GSC fate regulators

To explore the potential role of miRNAs in GSC fate, we obtained a collection of 90 miRNA mutant lines covering 111 *Drosophila* miRNAs from the Bloomington Drosophila Stock Center (S1 Table) [45]. We examined the ovaries of a collection of homozygous viable mutations affecting 54 miRNAs. An antibody to Hts (Hu li tai shao) labels the spectrosome organelle of GSCs and CBs and its branched derivative, the fusome, which connects sister cells of differentiating cysts derived from CBs. GSCs are evident from their anterior location and the presence of a spectrosome directly adjacent to a Cap cell. Significantly fewer GSCs were observed for the loss of function of miR-971, miR-975-976-977, miR-11, miR-219, or miR-999 (S1 Fig), suggesting a role in GSC maintenance. Defects in GSC or initial CB differentiation can be recognized by the accumulation of spectrosome-containing cells. Similar to previous findings, control germaria usually contained 2 or 3 GSCs and 1 CB to give an average of 3.36 ($n = 81$) spectrosome-containing cells per germarium (Figs 1B and S1A and S2 Table). We found that germaria with a loss of function of miR-970, miR-124, miR-986, or miR-957 showed an accumulation of undifferentiated germ cells with a spectrosome posterior to GSCs (Fig 1C and S2 Table). We measured the prevalence of the phenotype by scoring the percentage of germaria with 5 or more spectrosome-containing cells, and we referred to the additional cells posterior to normal GSC locations as CB-like cells. We found that approximately 50% ($n = 63$) of *miR-970*, 70% ($n = 70$) of *miR-124*, 65% ($n = 52$) of *miR-986*, and 83% ($n = 60$) of *miR-957* mutant germaria contained extra CB-like cells, while the incidence in the controls was less than 10% (Fig 1D). Of these miRNAs, miR-124 was of special interest because it is conserved among vertebrates and invertebrates [46–48] and has a number of known functions, including roles in *Drosophila* neuroblast proliferation, courtship, circadian locomotor, and rhythmic behaviors [49–52], as well as the inhibition of differentiation of mouse bone marrow mesenchymal stem cells and pre-cartilaginous stem cells [53,54].

### miR-124 expression in TF and Cap cells is critical for GSC daughter cell differentiation

We tested another 2 independent *miR-124* null allele lines (*miR-124[6]* and *miR-124[7]*) [49] and found similar phenotypes, with approximately 70% of germaria containing 5 or more cells

with spectrosomes in each case (Fig 1E–1G); the strength of this phenotype is similar to several prior studies of genotypes with impaired germline differentiation [28,41,55]. We also tested fertility. Both the number of eggs laid per day and the frequency of larval hatching per egg were significantly reduced in *miR-124* mutants relative to the $w^{1118}$ control (S2 Fig). To investigate the expression pattern of miR-124, we used a reporter line (*miR-124::dsRed*) for which RFP expression controlled by the promoter of miR-124 was previously found to mirror endogenous miR-124 expression in embryos [49]. We detected specific miR-124 expression in the anterior-most part of the germarium, and also in early follicle cells and some FSCs in region 2b but very little expression in between (Fig 1H and 1H'; quantitation in S3 Fig). The miR-124-positive cells in the anterior of the germarium mostly overlapped 2 cell types, TF and Cap cells, which both express the marker Lamin C (LamC) (yellow arrowheads in Fig 1H and 1H') [27,56]. There was also occasional expression in the most anterior ECs, immediately posterior to Cap cells (white arrowhead in Fig 1H and 1H').

To investigate whether miR-124 function is important in TF and Cap cells, we attempted to rescue the miR-124 loss-of-function differentiation defect through spatially selective expression of miR-124. We expressed *UAS-miR-124* specifically in TF and Cap cells using a *bab1-GAL4* line [57]. We first confirmed that expression of this *bab1-GAL4* line (bab$^{Agal4-5}$) is limited to TF and Cap cells and does not extend to ECs in adult ovaries (S4A and S4A' Fig). A different *bab1-GAL4* line, employed in some other studies, has a similar pattern but also has detectable expression in ECs [58]. The *bab1-GAL4* line we used is highly specific in adults but it is expressed in a broader group of somatic cells of the developing female gonad [57]. We therefore used a temperature-sensitive conditional GAL4/GAL80$^{ts}$ system to drive *UAS-miR-124* expression only in adults (S5A–S5B' Fig). Flies of appropriate genotypes were raised at 18˚C to suppress GAL4 activity during developmental stages. After eclosion, half of the adults were switched to 29˚C to inactivate GAL80$^{ts}$ and thereby permit GAL4 activation of *UAS-miR-124*, while the remaining half were kept at 18˚C. Germaria were examined 5 and 10 d later. The germaria of miR-124 mutant animals maintained at 18˚C throughout (*miR-124[6], GAL80$^{ts}$; bab1>miR-124* and *miR-124[7], GAL80$^{ts}$; bab1>miR-124*) contained extra CB-like cells at high penetrance (Fig 2A–2C), consistent with the absence of any potential rescue activity from the *UAS-miR-124* transgene. In contrast, when incubated at 29˚C for 5 d or 10 d as adults, most germaria contained the normal number of spectrosome-containing cells, indicating full phenotypic rescue (Figs 1D and 2A–2C). We also tested the converse temperature shift, raising flies at 29˚C and then moving half to 18˚C after eclosion. Flies raised and maintained at 29˚C throughout exhibited only a low basal frequency of excess CB-like cells (<20%) but this phenotype was increased to a penetrance of more than 65% by 5 d and more than 75% by 10 d at 18˚C for both *miR-124[6]* and *miR-124[7]* (S5C Fig). We conclude that miR-124 function is required in adult TF or Cap cells to promote GSC progeny differentiation.

We also performed the same type of experiment with *C587-GAL4* [23,26,27], which is specifically expressed in ECs but not TF or Cap cells (S4B and S4B' Fig). As expected, the germaria of animals maintained at 18˚C throughout (*C587; miR-124[6], GAL80$^{ts}$>miR-124* and *C587; miR-124[7], GAL80$^{ts}$>miR-124*) contained extra CB-like cells at high penetrance (Fig 2D). When incubated at 29˚C for 5 d or 10 d as adults, the germaria still contained an excess of spectrosome-containing cells (Fig 2D), albeit marginally fewer than at 18˚C. These results are consistent with miR-124 acting in TF or Cap cells, where miR-124 is strongly expressed, to promote GSC differentiation. It is possible that miR-124 activity in anterior ECs can also sometimes contribute to GSC differentiation, reflecting occasional expression in those cells.

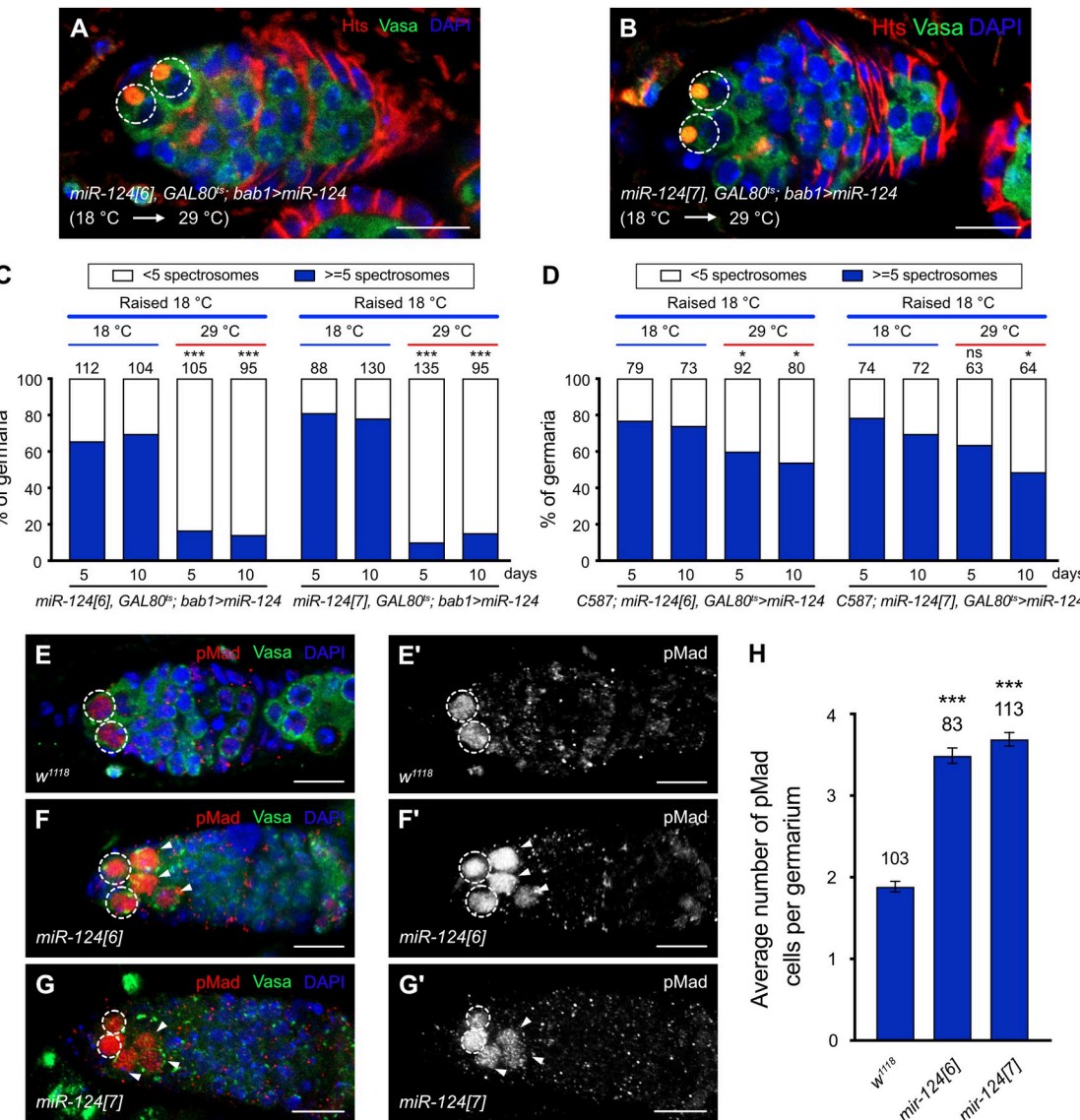

**Fig 2. miR-124 in GSC niche cells affects daughter cell differentiation.** (**A** and **B**) Ectopic expression of *UAS-miR-124* in the TF and Cap cells of *miR-124[6]* (**A**) and *miR-124[7]* (**B**) mutant backgrounds using *bab1-GAL4*. Germaria were immunostained for Hts (red), Vasa (green), and DAPI (blue). In (**A**) and (**B**), the GSCs are highlighted by dashed circles. Scale bar: 10 μm. (**C** and **D**) Animals carrying *UAS-miR-124* and *GAL80^{ts}; bab1-GAL4* (**C**) or *UAS-miR-124* and *C587-GAL4; GAL80^{ts}* (**D**) were raised at 18˚C up to eclosion and then maintained at 18˚C or 29˚C for the number of days indicated before ovary dissection. The percentage of germaria carrying 5 or more spectrosome-containing cells is shown, and the number of analyzed germaria is above each bar. Significance of 18˚C versus 29˚C for the same time period was determined by Fisher's exact two-sided test (* $P < 0.05$; *** $P < 0.001$; ns: not significant). (**E–G'**) Germaria from wild-type females (**E–E'**), *miR-124[6]* (**F–F'**), and *miR-124[7]* (**G–G'**) mutants were immunostained for a marker (pMad) of BMP pathway activity. pMad (red) in germline cells (marked by Vasa antibody, green) was normally present only in GSCs (**E–E'**, highlighted by dashed circle) but was present in additional cells (indicated by white arrowheads) in the germaria of *miR-124* mutants (**F–F'**, *miR-124[6]*; **G–G'**, *miR-124[7]*). (**E**), (**F**), and (**G**) show the merging of the 3 channels of pMad, Vasa, and DAPI (blue); (**E'**), (**F'**), and (**G'**) show pMad stained images in black and white. Scale bar: 10 μm. (**H**) Average numbers of cells with pMad staining per germarium in wild-type and 2 *miR-124* mutants. Left to right: $n$ = 103, 83, 113 biologically independent germaria. Data are presented as the mean ± SEM. Significance was analyzed by Kruskal–Wallis one-way ANOVA with Dunn's test (*** $P < 0.001$). The raw data underlying panels C, D, and H are available in S1 Data. BMP, bone morphogenetic protein; GSC, germline stem cell; TF, terminal filament.

## miR-124 limits BMP signaling in the CB differentiation niche

BMP signaling elevation is responsible for most or all CB differentiation defects caused by a variety of genetic alterations in ECs [26,55,59,60]. We examined BMP signaling in *miR-124* mutant germaria using phosphorylated Mad (pMad) as an indicator of BMP signaling activity. In GSCs, BMP signaling leads to the production of a strong pMad signal, whereas CBs and mitotic cysts show very little pMad signal [61]. In control germaria, pMad signal clearly above background levels was restricted to GSCs, with an average of 1.9 positive cells per germarium (Fig 2E, 2E', and 2H). In contrast, *mir-124[6]* and *mir-124[7]* mutant germaria included ectopic pMad staining in cells that were not adjacent to Cap cells, in addition to normal GSC staining (Fig 2F–2G' and quantified in S6 Fig). The average total number of pMad-positive cells was 3.5 (n = 83) for *mir-124[6]* and 3.7 (n = 113) for *mir-124[7]* (Fig 2H). The number of extra pMad-positive cells was slightly smaller than the increase of spectrosome-containing cells in both *mir-124[6]* and *mir-124[7]* germaria. Similar comparative results have been reported in almost all studies of germline differentiation defects and likely simply represent the different sensitivities of the 2 assays [23,27,62]. Accordingly, even among GSCs in control germaria, some do not have strong pMad staining. The total number of extra cells with a spectrosome or pMad staining was lower in *miR-124* mutant germaria than in several other situations where EC function was genetically compromised [27,63,64], and the extra cells were always observed close to the normal location of CBs. These observations suggest that the key deficit in the somatic cell environment of germline cells is confined to the anterior of the germarium and that differentiation is mostly delayed rather than arrested. This, in turn, suggests that the function of only the more anterior ECs is compromised by the absence of miR-124 in TF and Cap cells.

EC cell bodies reside next to the basement membrane around the circumference of the germarium, with cellular processes reaching into the interior to wrap developing germline cells. The presence of these long cellular processes is essential for germ cell differentiation [26,55,65]. Some perturbations in ECs that disrupt germline differentiation, such as the loss of Yorkie (Yki), also lead to the loss of EC processes [27]. We used *UAS-mCD8RFP* in combination with *C587-GAL4* to specifically label EC cellular processes with membrane-tethered RFP, mCD8RFP. Similar to the control germarium, EC cellular processes in both *mir-124[6]* and *mir-124[7]* mutant germaria spread normally into the germarium to ensheath differentiating germline cells (S7A–S7C' Fig). This contrasts with the response to expression of *UAS-yki RNAi* (S7D and S7D' Fig).

## Inhibition of Dpp production from ECs can rescue *miR-124* mutant differentiation defects

Genetic alterations of ECs have previously been found to increase BMP signaling by transcriptionally inducing the ectopic expression in ECs of either Dpp or Dally, a glycan that can promote the spread of Dpp signaling [66,67]. Here, the major focus of action of miR-124 is in TF and Cap cells rather than ECs and ectopic BMP signaling is limited to anterior regions. We therefore investigated whether Dpp or Dally production might be altered by miR-124, and in which cells this might occur.

To look for direct evidence of Dpp or Dally induction, we extracted RNA from germaria and early egg chambers of different genotypes for qRT-PCR tests (Fig 3A). We found that *dpp* RNA levels were 60% higher in *miR-124* mutant germaria than in controls, while *dally* RNA levels were unchanged (Fig 3B). We then used fluorescence-activated cell sorting (FACS) for the expression of *UAS-RFP* driven by *C587-GAL4* to purify ECs prior to qRT-PCR analysis (Fig 3A). We found that *dpp* RNA levels were increased almost 30-fold for *miR-124* mutants

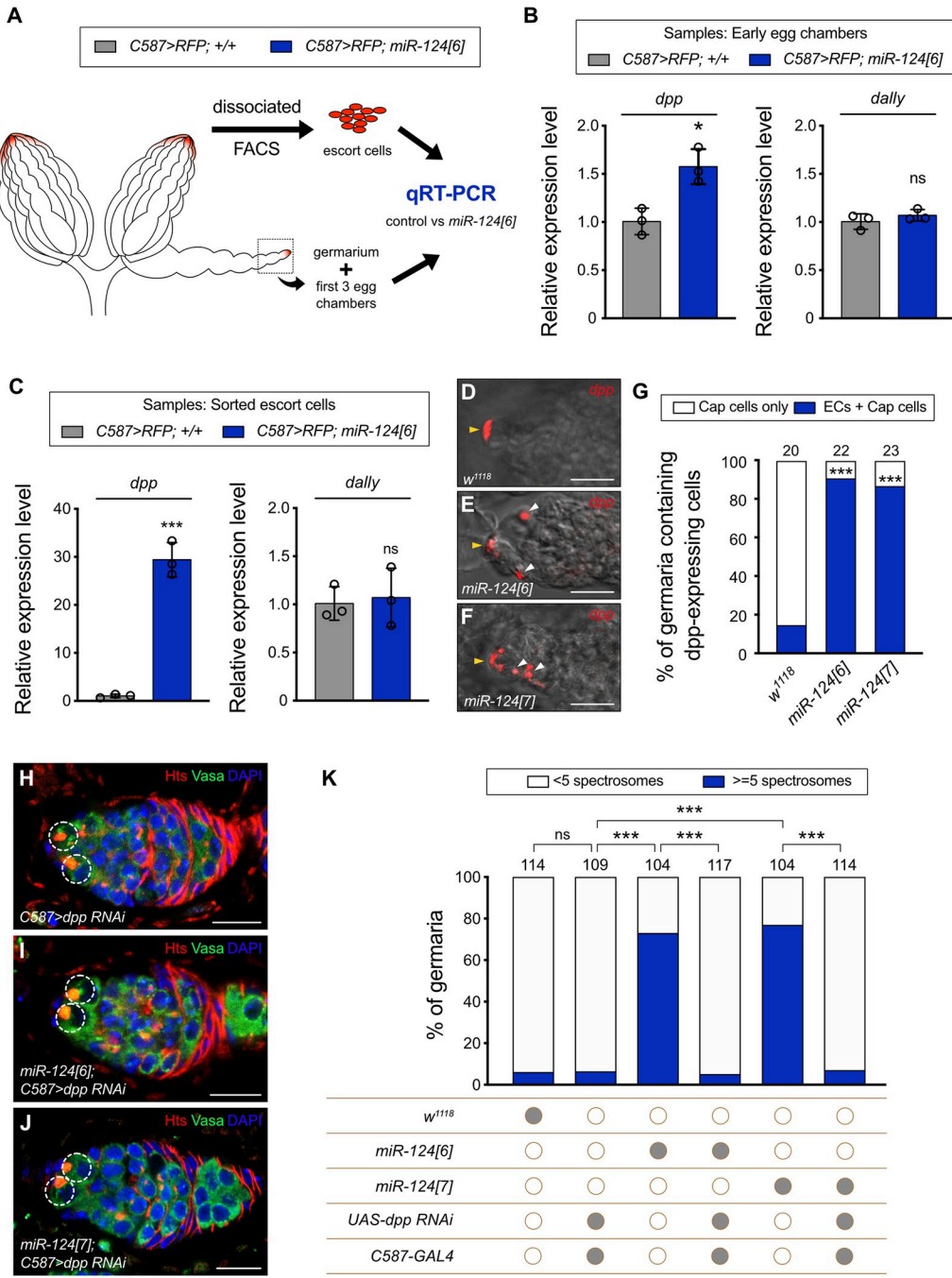

**Fig 3. Ectopic Dpp expression in ECs mediates germline differentiation defects in the miR-124 mutant. (A)** Scheme for collecting control and miR-124-deficient samples by dissociating and sorting RFP-labeled ECs (upper panel) or collecting the germaria of early egg chambers (lower panel). **(B)** Relative mRNA levels of *dpp* and *dally* in the collected germaria of the early egg chambers of *miR-124[6]* mutants compared with those of the control. Three biological replicates were performed. Data are presented as the mean value ± SEM; significance was determined by two-sided unpaired Student's *t* test (* $P < 0.05$; ns: not significant). **(C)** Relative mRNA levels of *dpp* and *dally* in the sorted ECs of *miR-124[6]* mutants compared with those of the control. Three biological replicates were performed. Data are presented as the mean value ± SEM; significance was determined by two-sided unpaired Student's *t* test (*** $P < 0.001$). **(D–F)** RNA in situ hybridization of *dpp* (red) was performed on germaria from wild-type females **(D)**, *miR-124[6]* **(E)**, and *miR-124[7]* **(F)**. *Dpp*-expressing Cap cells are marked by yellow arrowheads, and ECs with *dpp* signals are marked by white arrowheads. **(G)** Percentage of the germaria containing the *dpp* signal in either the Cap cells or ECs of wild-type (*w[1118]*) and 2 independent *miR-124* mutant lines, *miR-124[6]* and *miR-124[7]*. The number of analyzed germaria is shown above each bar. Significance between each *miR-124* mutant line and the control was determined by Fisher's exact

two-sided test (*** *P* < 0.001). **(H–J)** *C587-GAL4* was used to express *UAS-dpp RNAi* in ECs of wild-type **(H)**, *miR-124 [6]* **(I)**, or *miR-124[7]* **(J)** mutant. Germaria were stained for Hts (red), Vasa (green), and DAPI (blue) to label spectrosomes, fusomes, germline cells, and nuclei, respectively. In **(H–J)**, GSCs are highlighted by dashed circles. Scale bar: 10 μm. **(K)** Percentage of germaria carrying 5 or more spectrosome-containing cells with different genetic backgrounds. The number of analyzed germaria is shown above each bar. Significance between each transgenic group and the control group (*w*$^{1118}$) was determined by Fisher's exact two-sided test (*** *P* < 0.001; ns: not significant). Filled gray circles represent the presence, and empty brown circles represent the absence of a given transgene. The raw data underlying panels B, C, G, and K are available in S1 Data. Dpp, decapentaplegic; EC, escort cell; GSC, germline stem cell.

compared to controls, while the expression of *dally* was unchanged (Fig 3C). We conclude that *dpp* RNA is substantially induced in the ECs of *miR-124* mutant germaria.

We next examined the mRNA expression pattern of *dpp* directly by fluorescence in situ hybridization. Consistent with previous reports [28,29], *dpp* transcripts were detected with high specificity in the Cap cells of all wild-type germaria (Fig 3D). Approximately 15% of germaria (*n* = 20) also exhibited a *dpp* signal in 1 or more ECs (Fig 3G), with an average of 0.15 ECs in region 1 (the anterior portion of the germarium) and none in region 2a of control germaria (S4 Table). In contrast, in addition to a signal in Cap cells, 91% of the *mir-124[6]* mutant germaria (*n* = 22) and 87% of the *mir-124[7]* mutant germaria (*n* = 23) exhibited detectable *dpp* transcripts in ECs (Figs 3E–3G and S8). Among them, 1.05 ECs (*mir-124[6]*) and 1.09 ECs (*mir-124[7]*) on average in region 1 of mutant germaria, and 0.36 (*mir-124[6]*) and 0.48 (*mir-124[7]*) on average in region 2a of mutant germaria (S4 Table). Most of the ECs with ectopic *dpp* expression in region 1 were very close to the anterior of the germarium, in keeping with the location of ectopic germline pMad signals and extra CB-like cells.

We then tested the hypothesis that *miR-124* mutants delay germline differentiation by inducing ectopic Dpp expression in ECs using a genetic test. As reported previously [27], *dpp* and *dally* knockdown in adult ECs using *C587-GAL4* and *UAS-dpp RNAi* or *UAS-dally RNAi* had no significant effect on GSC maintenance or differentiation (Figs 3H and S9), allowing us to test their effect in combination with the loss of miR-124 function. We found that limiting *dpp* expression in ECs (*C587 > dpp RNAi*) completely suppressed the germline differentiation defect due to *miR-124* deletion, whereas the reduction of *dally* expression had no effect (Figs 3I–3K and S9). Thus, the loss of miR-124 induces a germline differentiation defect by causing ectopic Dpp expression in principally anterior ECs.

## miR-124 acts in GSC niche cells to prevent ectopic Dpp production in ECs

Our evidence thus far indicates that miR-124 acts in GSC niche cells to prevent a germline differentiation defect that is caused by ectopic Dpp production in ECs. These deductions rely on the specificity of *bab1-GAL4* for adult TF and Cap cells (and not ECs) and *C587-GAL4* specificity for ECs (and not TF or Cap cells). These specificities have been thoroughly demonstrated in normal ovaries through the direct examination of GAL4 reporters and a variety of genetic perturbation experiments (S4A–S4B' Fig) [27,28,68]. However, it is crucial to test also whether these specificities are retained in *miR-124* mutant germaria. For example, the loss of miR-124 might cause a full or partial EC-to-Cap cell transformation. To test this specific possibility, we used LamC to identify Cap cells in control, *mir-124[6]* and *mir-124[7]* mutant ovaries. In all cases, LamC expression was restricted to endogenous TF, Cap cells, and occasionally some muscle sheath cells but was not present in EC locations (S10A–S10C' Fig). Additionally, the total number of LamC-positive cells with normal appearance in Cap cell locations was unchanged in *mir-124[6]* and *mir-124[7]* mutant germaria (S10D Fig). These results indicate that there were no clear transformations between EC and Cap cells in *miR-124* mutant germaria. Equally important, the expression patterns of *bab1-GAL4* and *C587-GAL4* were

unchanged in *miR-124* mutant germaria, with no ectopic expression of *bab1-GAL4* in ECs or of *C587-GAL4* in Cap cells or TF cells (S11 Fig). Hence, we can be confident of the prior results that miR-124 functions in TF or Cap cells, which continue to function as GSC niche cells even in the absence of miR-124, to prevent ectopic Dpp production in ECs.

## *EGFR* is a potential direct target of *miR-124* in Cap cells

To understand how miR-124 can act in one cell type to affect another, we sought to identify its direct targets. MiRNAs usually bind to specific sequences within the 3′ UTRs of mRNAs, leading to degradation of the targeted mRNAs and reducing translation into proteins [69]. Several transcriptome studies in both vertebrates and invertebrates have, accordingly, demonstrated the up-regulation of direct targets upon miRNA depletion [70,71]. We therefore used *bab1--GAL4>UAS-RFP* animals to isolate Cap cells and TF cells from control and mutant ovaries using FACS and compare RNA profiles (Fig 4A). RNA sequencing reads were aligned to the *Drosophila* genome, and gene expression levels were quantified using the RSEM pipeline. Among the 130 predicted *miR-124* targets (TargetScan Fly, http://www.targetscan.org/), we found a group of 57 genes with increased expression in *miR-124* mutant Cap and TF cells (Fig 4B and S3 Table). The *EGFR* gene attracted our attention due to the magnitude of the increase (ranking first). The EGFR signaling pathway has previously been implicated in germline differentiation, but acting positively in ECs [72].

There is a single miR-124 canonical recognition site (CACGGAA) located within the *EGFR* 3′ UTR (Fig 4C). To test whether this site can mediate the effects of *miR-124* on *EGFR* expression, we performed dual luciferase reporter assays in HEK293 cells (Fig 4C). The 3′ UTR of *EGFR* (UTR$^{WT}$) was cloned into the pMIR-REPORT vector downstream of a firefly luciferase gene, and mutations were introduced into the 3′ UTR (UTR$^{MUT}$) to abolish the miRNA target site as a negative control. Compared to the control miRNA sequences (NC, EXIQON 479903–001, USA), the luciferase reporter activity of the *EGFR*-3′ UTR$^{WT}$ construct was significantly reduced in the presence of *miR-124*; moreover, the altered *EGFR*-3′ UTR$^{MUT}$ construct abolished the repression (Fig 4D). A similar test was performed in *Drosophila* S2 cells, directly measuring RNA levels of complete EGFR mRNAs. The qRT-PCR results showed that the transcript levels of *EGFR* were significantly decreased by miR-124 in *EGFR*$^{WT}$ co-transfected cells, but were not changed by miR-124 in *EGFR*$^{MUT}$ co-transfected cells (S12 Fig). To investigate whether the target site of miR-124 in *EGFR* 3′ UTR is conserved, we searched the *EGFR* 3′ UTR sequences in 20 insect species representing 6 orders, including Diptera (13), Lepidoptera (2), Hymenoptera (2), Coleoptera (1), Hemiptera (1), and Orthopterodea (1). The alignment results showed that the target site of miR-124 in the *EGFR* 3′ UTR is conserved in many *Drosophila* species, but is not evident in other insects (S13 Fig). These data confirm that miR-124 can act at a specific site within the 3′ UTR to reduce *EGFR* expression, and this action is likely conserved among several *Drosophila* species.

If EGFR is a significant target for miR-124 in TF and Cap cells, we would expect to see increased EGFR expression or activity in *miR-124* mutants. Consistent with previous findings [26], we found a strong phosphorylated ERK (pERK) MAP kinase signal, indicative of EGFR signaling pathway activity, in ECs but little or no signal in the TF or Cap cells of normal germaria (Fig 4E and 4E'). However, in *miR-124* mutant germaria, there was clearly an increased pERK signal in Cap cells (Fig 4F–4G'). When we expressed *UAS-miR-124* in TF and Cap cells in a *miR-124* mutant background, the increased pERK signal was rescued to the level in Cap cells of normal germaria (Fig 4H and 4H'). We next measured the average intensity of pERK staining among at least 3 Cap cells and among at least 10 ECs in each germarium. We found a significant elevation of pERK in the Cap cells of *miR-124* mutant germaria, while the pERK in

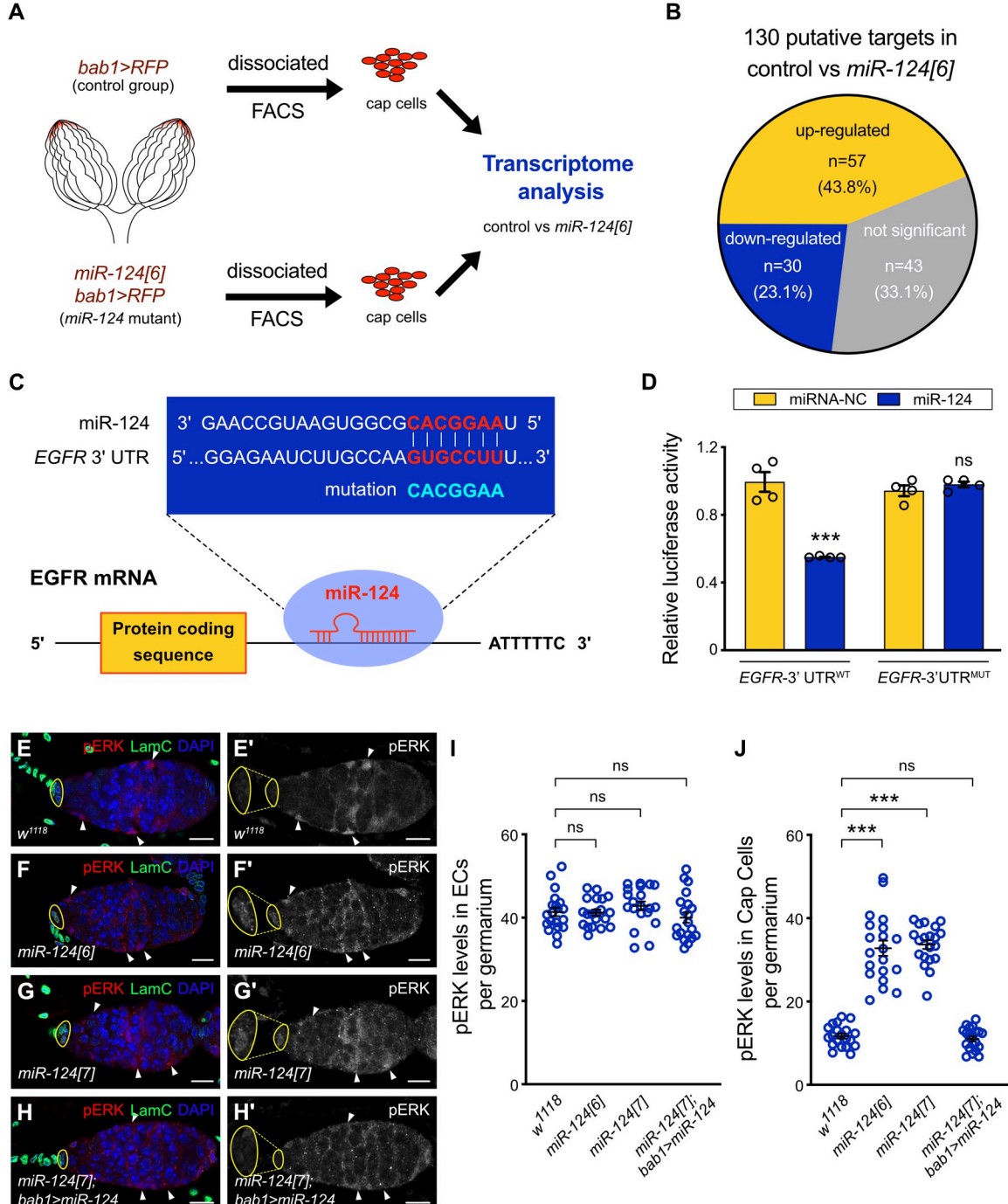

**Fig 4. *EGFR* is a target of miR-124 in Cap cells. (A)** Scheme for dissociating and sorting RFP-labeled control and miR-124-deficient Cap cells (plus TF cells). **(B)** Differential expression of the 130 predicted miR-124 targets from the sorted cells of the *miR-124[6]* mutant vs. control. Among them, 43.8% (57 out of 130, orange part) showed higher expressions. **(C)** The potential binding site of *miR-124* in the 3′ UTR of *EGFR* is presented. Nucleotide sequences showed the complementarity of *miR-124* with the wild-type (WT) or mutant (MUT) site (blue letter) in *EGFR* on the *miR-124* seed-binding region. **(D)** The *pMIR-REPORT-EGFR* luciferase construct without (*EGFR*-3′ UTR$^{WT}$) or with (*EGFR*-3′ UTR$^{MUT}$) a mutation site was cotransfected into HEK293 cells with either a negative control miRNA (miRNA-NC) or miR-124, and the corresponding luciferase activity was determined by a dual-luciferase assay. Four biological replicates were performed. Data are presented as the mean ± SEM; significance was analyzed by two-way ANOVA with Sidak's multiple comparisons test (*** $P < 0.001$; ns: not significant). **(E–H')** Germaria from wild-type females (**E–E'**), *miR-124[6]* (**F–F'**), *miR-124[7]* (**G–G'**) mutants, and *miR-124[7]; bab1 > UAS-miR-124* (**H–H'**) were immunostained for an indicator (pERK) of EGFR-MAPK signaling activity. pERK (red) was highly expressed in the ECs (marked by arrowheads) of these different genotypes. In contrast, the

pERK in the Cap cells (marked by yellow flat circle and stained for LamC antibody in **(E)**, **(F)**, **(G)**, and **(H)**, green) was undetectable for control females **(E–E')** but was strongly expressed in 2 *miR-124* mutants **(F–G')**. Overexpression of miR-124 activity (*bab1 > UAS-miR-124*) in Cap cells largely suppressed the increased pERK signal in *miR-124[7]* mutant **(H–H')**. **(E)**, **(F)**, **(G)**, and **(H)** show the merging of the 3 channels of pERK, Vasa, and DAPI (blue); **(E')**, **(F')**, **(G')**, and **(H')** show pERK stained images in black and white. Scale bar: 10 μm. **(I)** and **(J)** Quantitation of *EGFR* levels in ECs **(I)** or Cap cells **(J)** per germarium for the 4 genotypes in **(E–H')**. Each plot in **(I)** indicates the mean intensity from 10 ECs for each germarium; each plot in **(J)** indicates the mean intensity of at least 3 Cap cells within each germarium (*n* = 20 *Drosophila* germaria were examined for each group). Data are presented as the mean ± SEM. Significance was determined by Kruskal–Wallis one-way ANOVA with Dunn's test (*** $P < 0.001$; ns: not significant). The raw data underlying panels D, I, and J are available in S1 Data. EGFR, epidermal growth factor receptor; TF, terminal filament; UTR, untranslated region.

ECs appeared unchanged (Fig 4I and 4J). We deduce that miR-124 normally limits EGFR pathway activity in Cap cells and likely does so by directly interacting with the EGFR 3′ UTR to reduce expression.

## Altered EGFR signaling as a mediator of miR-124 function in Cap cells

To determine whether EGFR might be a functionally important target for miR-124 in ovaries, we first determined the consequences of overexpressing EGFR in TF and Cap cells by using the combination of *UAS-EGFR* and *bab1-GAL4* compared to controls with each genetic element alone. We found that EGFR overexpression in TF and Cap cells caused a significant germline differentiation defect, with approximately 70% of germaria containing more than 5 cells with a spectrosome (Fig 5A–5C). Similar results were found when we used a temperature-sensitive conditional GAL4/GAL80^ts system to drive *UAS-EGFR* expression only in adults (S14 Fig). We also found that *EGFR* overexpression in TF and Cap cells induced ectopic pMad staining, with significantly more pMad-positive cells than in controls (S15 Fig). The locations and magnitudes of the changes in pMad staining and spectrosome-containing cells were very similar for the loss of miR-124 and overexpression of *EGFR*. *EGFR* overexpression in TF and Cap cells also significantly increased *dpp* RNA levels in RNA isolated from the germarium and early egg chambers (Fig 5D). The increase in *dpp* RNA was 90%, similar to the 60% increase found in analogous *miR-124[6]* samples that were not enriched for ECs. We next determined the consequences of overexpressing EGFR in ECs by using the combination of *UAS-EGFR* and *C587-GAL4*. We found that EGFR overexpression in ECs caused no change in germline differentiation (S16 Fig).

Thus, the *EGFR* gene includes a functional target site for miR-124, *EGFR* RNA levels were increased in TF and Cap cells by the loss of miR-124, and the overexpression of EGFR in TF and Cap cells, the critical site of action of miR-124, produced phenotypes very similar to those of the loss of miR-124, including *dpp* RNA induction, ectopic BMP signaling, and defective germline differentiation.

To test whether the *miR-124* mutant differentiation defect depends on elevated EGFR activity in TF and Cap cells, we reduced EGFR expression in those cells using *bab1-GAL4* and *UAS-EGFR RNAi* in normal and *miR-124* mutant germaria. The reduction of EGFR alone in TF and Cap cells had no effect on germarial appearance or germline differentiation, consistent with the observed very low levels of EGFR signaling pathway activity in these cells. However, the reduction of EGFR expression in TF and Cap cells fully suppressed the differentiation defects of *miR-124* mutant animals (Fig 5E–5H). These results indicate that miR-124 normally restricts EGFR expression, likely through direct interaction with the EGFR 3′ UTR, to limit EGFR signaling in TF and Cap cells, thereby preventing ectopic Dpp expression in ECs and delayed germline differentiation.

## Notch signaling acts as a connection between the 2 different niches

To find out how the miR-124-EGFR cascade in TF and Cap cells alters *dpp* transcription in ECs, we searched for potential mediators using the same set of RNASeq results that identified

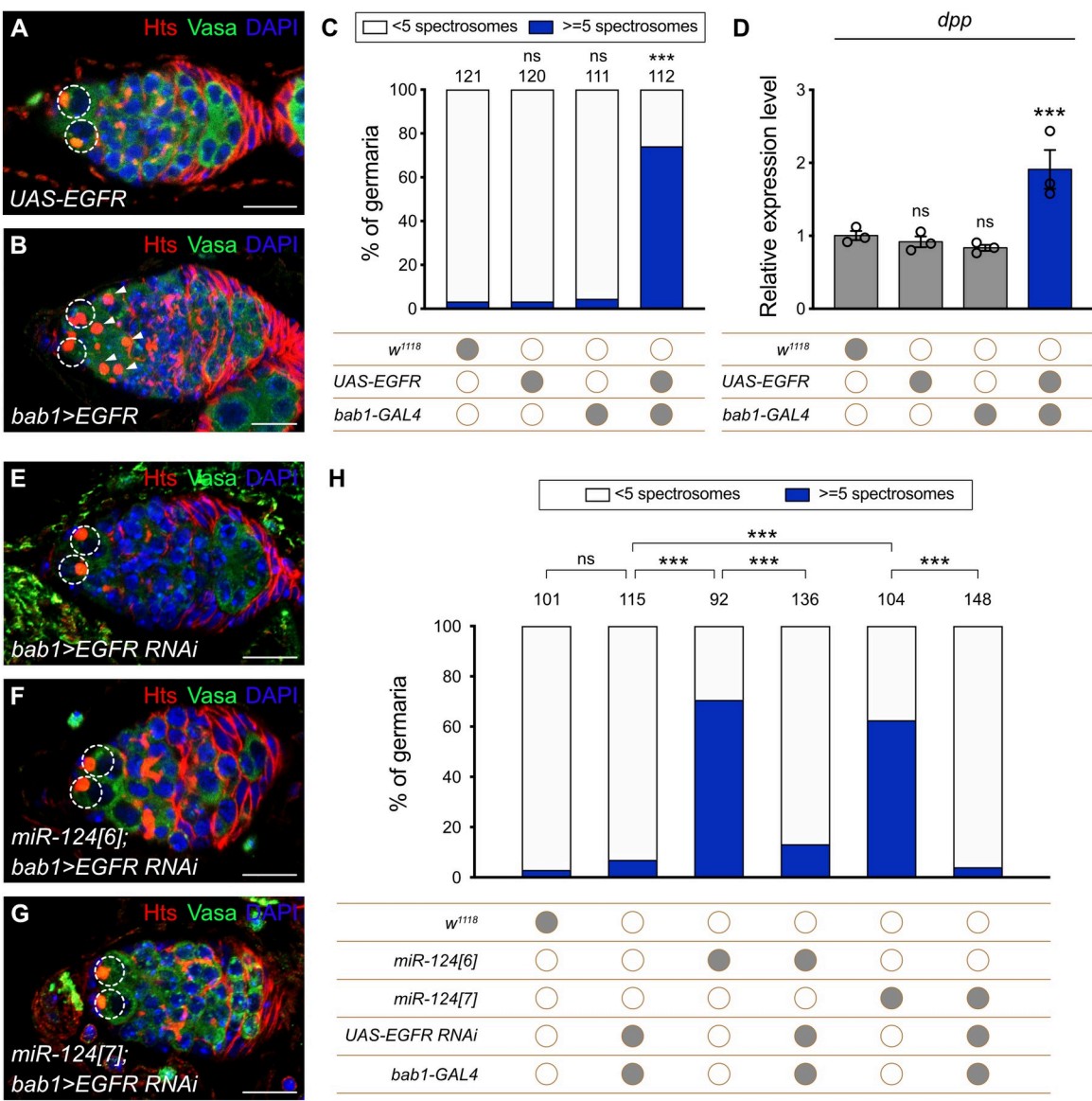

**Fig 5. GSC progeny differentiation depends on limiting EGFR signaling activity in Cap cells.** (**A** and **B**) Germaria of animals with *UAS-EGFR* alone (**A**) or *bab1*-GAL4 plus *UAS- EGFR* (**B**) were stained for Hts (red), Vasa (green), and DAPI (blue), respectively. In (**A**) and (**B**), the GSCs are highlighted by dashed circles, whereas the CBs or CB-like in (**B**) are indicated by arrowheads. Scale bar: 10 μm. (**C**) Percentage of germaria carrying 5 or more spectrosome-containing cells with different genotypes. The number of analyzed germaria is shown above each bar. Significance was determined by Fisher's exact two-sided test (*** $P < 0.001$; ns: not significant). (**D**) Relative mRNA levels of *dpp* when EGFR was overexpressed in Cap cells. Three biological replicates were performed. Data are presented as the mean ± SEM; significance was determined by Kruskal–Wallis one-way ANOVA with a Dunn's test (*** $P < 0.001$; ns: not significant). In (**C**) and (**D**), filled gray circles represent the presence, and empty brown circles represent the absence of a given transgene. (**E–G**) *bab1-GAL4* was used to express *UAS-EGFR RNAi* in wild-type (**E**), *miR-124[6]* (**F**), or *miR-124[7]* (**G**) mutant Cap cells. Germaria were stained for Hts (red), Vasa (green), and DAPI (blue). GSCs are highlighted by a dashed circle in (**E**), (**F**), and (**G**). Reduction of EGFR in Cap cells restored the normal germ cell differentiation for *miR-124* mutant lines. Scale bar: 10 μm. (**H**) Percentage of the germaria carrying 5 or more spectrosome-containing cells with different genotypes. The number of analyzed germaria is shown above each bar. Significance between each transgenic group and the control group (*w[1118]*) was determined by Fisher's exact two-sided test (*** $P < 0.001$; ns: not significant). Filled gray circles represent the presence, and empty brown circles represent the absence, of a given transgene. The raw data underlying panels C, D, and H are available in S1 Data. CB, cystoblast; Dpp, decapentaplegic; EGFR, epidermal growth factor receptor; GSC, germline stem cell.

regulation of EGFR by miR-124. We found that the expression of Delta (Dl), the ligand of Notch signaling, was significantly increased in *miR-124* mutant cells using the exact test of edgeR based on RNA-seq reads count (TPM = 1.97 in CK, TPM = 20.82 in mutant; $P < 0.001$) (S17 Fig). Because Dl is a target of EGFR signaling in some settings [73,74], we then asked whether Notch signaling mediates the connection between the miR-124-EGFR cascade in Cap cells and *dpp* expression in ECs. Consistent with previous findings [75], we found little or no Dl protein expression in Cap cells of normal germaria (Fig 6A and 6A'). However, in miR-124 mutant germaria, there was clearly an increased Dl signal in Cap cells (Fig 6B, 6B', and 6E). Moreover, reducing EGFR expression in Cap cells significantly suppressed the increased Dl signal of *miR-124* mutant animals (Fig 6C, 6C', and 6E). In addition, overexpression of EGFR alone sufficed to increase Dl expression in Cap cells (Fig 6D, 6D', and 6E). Collectively, these results suggest that Dl is a downstream target of the miR-124-EGFR cascade in Cap cells.

To test whether the ectopic *dpp* expression in ECs of *miR-124* mutant depends on elevated Dl protein in Cap cells, we reduced Notch signaling activity in ECs using *C587-GAL4* and *UAS-Notch RNAi* in normal and *miR-124* mutant germaria. The reduction of Notch activity alone in ECs had no effect on germline differentiation. However, the reduction of Notch expression in ECs significantly suppressed the differentiation defects of *miR-124* mutant animals (Fig 6F–6H). Accordingly, reduction of Notch activity in ECs also significantly decreased *dpp* RNA levels (Fig 6I). These results support the interpretation that Notch signaling acts directly as a relay between the miR-124-EGFR cascade in Cap cells and *dpp* transcripts in ECs, leading to a defect in germline differentiation in the absence of miR-124 (Fig 6J). We want to note that the most prevalent sites of ectopic *dpp* were in anterior ECs (S4 Table), which can contact Cap cells and the inside Notch signaling directly. The less frequent activation of *dpp* in more posterior ECs (S4 Table) might be due to longer-range Dl-Notch signaling [76,77], potentially mediated by EC processes, or to occasional ectopic Dl production in anterior ECs.

## Discussion

Cell behavior is universally guided by the microenvironment. The microenvironment of an adult stem cell must regulate proliferation and must prevent differentiation [78,79]. However, differentiation must commence outside that environment. A unitary decision of whether to differentiate or not could, in principle, be guided by a single, simple, highly localized signal. However, analysis of different stem cell paradigms is beginning to reveal far more complexity. This motivates a full understanding of the molecular details of a stem cell niche, potentially revealing benefits of more complex regulation. Only a few paradigms, including *Drosophila* GSCs, are currently well-suited to intensive, thorough interrogation [80–83].

*Drosophila* GSCs provide an example of a type of stem cell that is maintained by generally long-lived stem cells, which repeatedly divide to yield a stem cell and a non-stem cell daughter (single-cell asymmetry). Arguably, normal maturation of cysts in the germarium towards encapsulation and egg chamber production is best supported if non-stem GSC daughters start to differentiate immediately and irreversibly. That requires an abrupt signaling change between 2 very closely spaced cells (a GSC and a CB daughter). Since only GSCs contact a specific specialized cell type (Cap cells), such a signal could hypothetically be contact-dependent. The key signal produced by Cap cells for GSC maintenance is a BMP ligand, which can potentially travel beyond its source [10,26,67]. Possibly, the reason for the choice of a potentially long-range signal centers on resilience. Each niche contains only 2 to 3 GSCs, so it may be important to quickly replace any GSC that inadvertently loses contact with Cap cells, or to permit more drastic de-differentiation under conditions of stress that lead to loss of all GSCs.

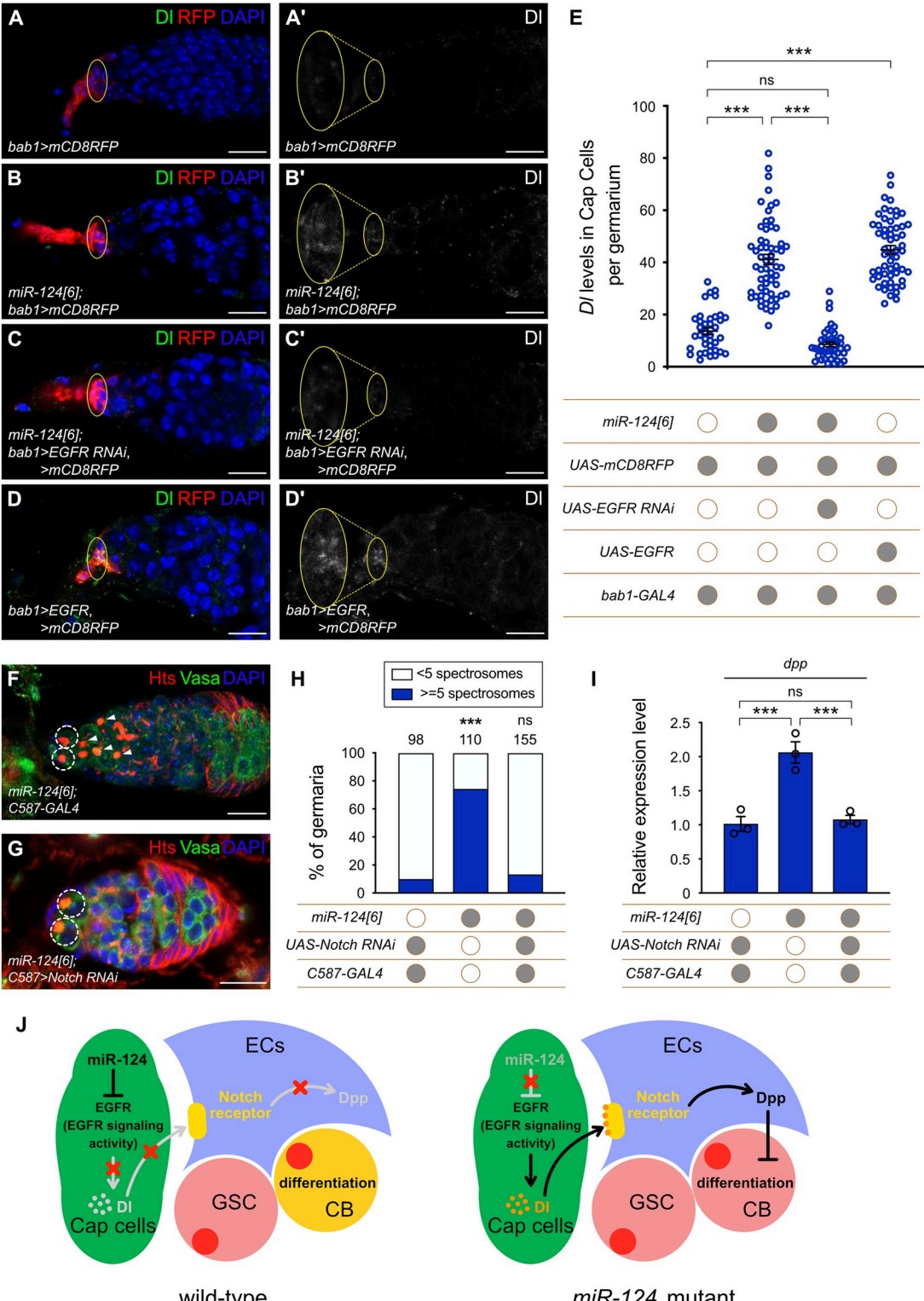

**Fig 6. Notch signaling acts as a connection between EFGR activity in Cap cells and *dpp* transcripts in ECs. (A–D')** Germaria from wild-type females **(A–A')**, *miR-124[6]* **(B–B')**, *miR-124[6]; bab1 > UAS-EGFR RNAi* **(C–C')**, and *bab1 > UAS-EGFR* **(D–D')** were immunostained for Dl (the ligand of Notch signaling). Dl (green) showed extremely low expression in Cap cells, marked by yellow flat circle and with mCD8RFP in control females **(A–A')** but was more strongly expressed in *miR-124* mutants **(B–B')** and *EGFR*-overexpressing germaria **(D–D')**. In contrast, reduction of EGFR activity (*UAS-EGFR RNAi*) in Cap cells

largely suppressed the increased Dl signal in *miR-124* mutants **(C–C')**. **(A)**, **(B)**, **(C)**, and **(D)** show the merging of the 3 channels of Dl, RFP, and DAPI (blue); **(A')**, **(B')**, **(C')**, and **(D')** show Dl stained images in black and white. Scale bar: 10 μm. **(E)** Quantitation of Dl protein levels in Cap cells per germarium of flies with different genotypes in **(A–D')**. Each plot indicates the mean intensity from 3 Cap cells within each germarium (left to right: $n$ = 38, 59, 42, and 57 were examined for each group). Data are presented as the mean ± SEM. Significance was determined by Kruskal–Wallis one-way ANOVA with Dunn's test (*** $P < 0.001$; ns: not significant). **(F and G)** *C587-GAL4* was used alone to express *UAS-Notch RNAi* **(G)** in ECs of the *miR-124[6]* mutant. Germaria of both genotypes were stained for Hts (red), Vasa (green), and DAPI (blue), respectively. In **(F)** and **(G)**, the GSCs are highlighted by dashed circles, whereas the CBs or CB-like are indicated by arrowheads. Scale bar: 10 μm. **(H)** Percentage of germaria carrying 5 or more spectrosome-containing cells with different genotypes. The number of analyzed germaria is shown above each bar. Significance was determined by Fisher's exact two-sided test (*** $P < 0.001$; ns: not significant). **(I)** Relative mRNA levels of *dpp* in the collected germaria of the early egg chambers from different *Drosophila* in **(H)**. Three biological replicates were performed. Data are presented as the mean value ± SEM; significance was determined by ordinary one-way ANOVA along with Holm–Sidak's test (*** $P < 0.001$; ns: not significant). **(J)** Schematic diagram showing that miR-124 non-cell-autonomously maintains the GSC daughter cell differentiation niche through the EGFR and Notch signaling pathways. In wild-type females (left panel), miR-124 directly down-regulates EGFR in Cap cells, which suppresses Dl expression, thus limiting Notch activity in ECs to prevent BMP ligand production, thereby providing a permissive environment for GSC progeny differentiation. In *miR-124* mutants (right panel), higher EGFR activity in Cap cells elevates Dl production, which activates Notch activity to increase Dpp in ECs and cause germline differentiation defects. The raw data underlying panels E, H, and I are available in S1 Data. BMP, bone morphogenetic protein; CB, cystoblast; Dpp, decapentaplegic; EC, escort cell; EGFR, epidermal growth factor receptor; GSC, germline stem cell.

To accommodate the use of a ligand that signals over large distances in other tissues, such as the wing imaginal disc, a large number of mechanisms have evolved to limit BMP signaling in the germarium [23–27]. These active mechanisms involve ECs, the somatic cells posterior to Cap cells and GSCs, so that ECs are considered a differentiation niche. Here, we report that these 2 contrasting niches interact. Specifically, we found that a microRNA, miR-124, is required in the adult GSC niche (Cap cells) to limit EGFR expression, which non-cell-autonomously prevents BMP ligand production in the differentiation niche (ECs), thereby promoting GSC progeny differentiation. The net effect of this interaction is to sharpen the border between stem cell and differentiation niches.

Our findings raise numerous questions. First, why is it necessary for miR-124 to limit EGFR activity by limiting EGFR protein in GSC niche cells? It has previously been shown that EGFR activity in ECs is required to suppress *dally* transcription and hence limit the posterior spread of BMP signaling to allow germline differentiation [59]. EGFR activity is stimulated by multiple ligands produced by germline cells, especially GSCs. Thus, EGFR ligands are produced at significant levels close to Cap cells to serve a functional purpose in ECs. This circumstance inevitably could lead to significant EGFR activation in Cap cells. Our findings show that a specific mechanism is in place to limit EGFR activity in TF and Cap cells, namely, a posttranscriptional reduction of EGFR RNA initiated by selective expression of miR-124. We do not know what controls the pattern of miR-124 expression or why EGFR expression is modulated between Cap cells and ECs in this manner, rather than by direct transcriptional control of the EGFR gene. It is certainly interesting to note how intricately EGFR activity modulation is used to restrict BMP signaling range: EGFR activity must be present in ECs to restrict *dally* expression, while EGFR activity must be limited in Cap cells to restrict *dpp* expression in ECs. The EGFR-Dally regulatory circuitry also acts during ovary establishment so that primordial germ cells adjacent to Cap cells become GSCs, while those adjacent to EC precursors, known as intermingled cells, start to differentiate during pupal development [72,84].

A second question is how EFGR activity in Cap cells connects to *dpp* expression in ECs. In this study, we found the expression of Dl in Cap cells was significantly up-regulated by overexpression of EGFR in the same cells or by loss of miR-124. In the latter scenario, increased Dl was suppressed by reducing EGFR expression in TF and Cap cells. Thus, in the absence of miR-124, ectopic Dl from Cap cells may activate Notch in nearby ECs. Accordingly, ectopic *dpp* expression and impaired germline differentiation in *miR-124* mutants were significantly

rescued by blocking Notch activity in ECs. Our findings therefore suggest that Notch signaling acts as a connection between EFGR activity in Cap cells and *dpp* expression in ECs. Notch signaling generally involves direct apposition of Notch-expressing cells with ligand-expressing cells. The majority of directly visualized alterations (ectopic *dpp* RNA, ectopic pMad and ectopic cells with spectrosomes) consequent to loss of miR-124 or EGFR overexpression was in anterior ECs, very close to Cap cells. However, there clearly were also alterations in a few ECs in more posterior locations. There are several possibilities, which remain to be investigated, for this longer distance effect. These include a possible direct connection to Cap cells through long cellular processes [85], an indirect relay from ECs with elevated Notch signaling, mediators other than Dl-Notch signaling, EC movement towards and away from Cap cells over time, or an occasional initiation of the whole cascade from anterior ECs expressing miR-124 and behaving somewhat similarly to Cap cells.

Interestingly, Dl to Notch signaling is an essential step for specifying Cap cells during ovary development and in the adult, continued signaling, originating from germline cells, is required for Cap cell maintenance, while excessive Notch activation may cause ECs to behave like Cap cells [75,86]. Further study showed that TFs, the first ovarian cells to be specified in larvae, initially all have active Notch signaling. However, an ecdysone signal then leads to the expression of miR-125 specifically in the most posterior TF, converting this cell into a Dl-producing cell, which then signals to more posterior cells to become Cap cells [87]. Thereafter, continued Notch activity in Cap cells would contribute to these cells not producing Dl ligand due to the universal negative feedback between Notch activity and Notch ligand expression. Altogether, these studies suggest Notch signaling in adult ECs, from the sum of germline and somatic cell ligands, is deliberately kept low to allow normal EC function. We have discovered that this state of Notch ligand limitation is re-inforced in adult ovaries by the expression of miR-124, which limits the potential for Dl induction via EGFR activation in Cap cells. Thus, key steps leading to both the establishment and the maintenance of a tightly localized niche for GSCs depend on a similar interplay between a specific microRNA expressed in a restricted spatial pattern and Notch ligand production.

A third question relates to the general roles of microRNAs. We have found that the phenotype of loss of miR-124 is reproduced at similar penetrance and severity by overexpression of EGFR in TF and Cap cells. Moreover, the miR-124 loss of function phenotype was fully suppressed by reducing EGFR expression in TF and Cap cells. Those results suggest that altered EGFR expression is the sole key response to loss of miR-124 that has functional consequence for germline cell differentiation. miR-124 can act directly on EGFR RNA in heterologous cells and likely therefore acts directly on EGFR RNA in TF and Cap cells. It is sometimes found that an microRNA affects several components of a functional unit, such as multiple members of a signal transduction pathway or components affecting cell growth, in order to achieve a cumulative effect [88]. However, there are relatively few rigorous genetic tests of such indications. Conversely, however, there are some clear examples of genetic tests similar to those we have employed, which identify single microRNA targets as key effectors [89–93]. Thus, our identification of EGFR as a principal singular direct effector for miR-124 in TF and Cap cells may represent the norm. Delta induction was found to be a key consequence of increased EGFR activity but Delta RNA has no sequence motifs consistent with being a potential direct miR-124 target. Finally, some microRNAs have conserved targets among divergent organisms or are thought to have functions associated with specific cell types or physiological activities. miR-124 has previously been associated with cells of the central nervous system in many animals, including *Drosophila*, albeit with different effects and direct targets in different settings [49–52]. Clearly, evolution has exploited the opportunity for using a specific miRNA also for odd jobs elsewhere, as illustrated here for miR-124 in the ovary. miR-124 has not previously

been connected to EGFR in *Drosophila* but in mammals there are reports of numerous connections, in both directions, between EGFR signaling and miR-124 expression [94,95]. Perhaps an existing potential regulatory connection between miR-124 and EGFR is one reason why this particular interaction was selected for down-regulation of EGFR in Cap cells.

It is already clear that miRNAs can play important roles in stem cell maintenance and differentiation in various settings [31,38,42,96]. Aside from the role of miR-124, we also found that miR-970, miR-986, and miR-957 are important for GSC progeny differentiation, while miR-971, miR-975-976-977, miR-11, miR-219, and miR-999 are necessary for GSC maintenance. Overall, more than 20% of the tested miRNA mutants were found to be involved in germline stem cell maintenance and differentiation. Interestingly, other research groups have identified additional roles for miRNAs in both adult and developing *Drosophila* germline stem cell niches. The mir-310 cluster responds to diet to regulate the strength of Hedgehog signaling [97]. miR-125 and let-7 are involved in steps preceding the specification of Cap cells in larval ovaries, as explained above [31,87]. All of these studies suggest that we have barely scratched the surface of the highly regulated interactions that lead to the production and maintenance of closely apposed stem cell and differentiation niches of carefully orchestrated strengths and spatial reaches.

## Materials and methods

### *Drosophila* stocks

*Drosophila* stocks were maintained on standard cornmeal/molasses/agar medium within 6-ounce square bottom plastic fly bottles at 25˚C. All experiments with wild-type *D. melanogaster* were carried out with the $w^{1118}$ strain. A total of 90 miRNA mutant lines covering 111 *Drosophila* miRNAs were obtained from Bloomington *Drosophila* Stock Center (BL) (S1 Table). Other strains were also used in our previous study: *C587-GAL4*, *tub-GAL80^ts^*, *UAS-mCD8-RFP*, and *UAS-RFP* [27]. *bab1-GAL4* line (BL#6802) was provided by Prof. Zheng Guo (Huazhong University of Science and Technology, China) [57]. *UAS-miR-124* (BL#41126), *UAS-dpp RNAi* (BL#31530), *UAS-dally RNAi* (BL#33952), *UAS-EGFR* (BL#9535), *UAS-EGFR RNAi* (BL#31183), UAS-*Notch RNAi* (BL#7078) lines were also obtained from BL. Two independent knockout alleles, *miR-124[6]* and *miR-124[7],* and the reporter line *miR-124:dsRed* were provided by Prof. Eric C. Lai (Sloan-Kettering Institute, New York, New York, United States of America). Characterization of miR-124 loss-of-function alleles in this study is provided in S18 Fig, and their original resources also can be found in Sun and colleagues [49] and Weng and Cohen [50].

### Immunohistochemistry

*Drosophila* ovaries were dissected in 1 × PBS, fixed in 4% paraformaldehyde in PBS for 15 min, rinsed 3 times with 1 × PBST (PBS containing 0.1% Triton X-100 and 0.05% Tween 20), blocked with 1% bovine serum albumin in 1× PBST, and stained with primary antibodies overnight at 4˚C. The following antibodies were used: mouse anti-Hts 1B1 [1:20; Developmental Studies Hybridoma Bank (DSHB)], rat anti-Vasa (1:10; DSHB), mouse anti-Lamin C LC28.26 (1:50; DSHB), mouse anti-Delta (1:100; DSHB), rabbit anti-pS423/S425 Smad3 (1:100; Abcam, Cat#52903), rabbit anti-pT202/Y204 ERK1/2 (1:200; Cell Signaling Technology, Cat#4370S). The ovaries were then washed 3 times in 1 × PBST and incubated with Alexa Fluor 488 or 594 secondary antibodies (1:1,000; Molecular Probes) for 2 h at room temperature, followed by 2 washes in 1 × PBST and 1 wash in 1 × PBS and mounting in ProLong Gold Antifade Mountant with DAPI (Invitrogen, Cat#P36935). Fluorescence images were captured using 63 ×/1.4 NA oil-immersion lenses with a Zeiss LSM 800 confocal microscope and then processed using Affinity Designer (Affinity) or ImageJ (National Institutes of Health).

### Drosophila fertility assay

Newly emerged male and female *w*<sup>1118</sup> or *miR-124* mutant (*miR-124[6]* and *miR-124[7]*) adults were collected. Twenty 2-day-old females and five 2-day-old males of each *Drosophila* line were placed into a fly bottle, which covered a 3.5-cm diameter Petri dish containing fly food medium. The food dishes were replaced daily for a period of 5 days. The total number of eggs deposited on the dishes each day was counted, and the hatching larvae were counted 2 days after egg collection. Egg hatching rate was calculated as the total number of hatching larvae divided by the number of total eggs laid.

### Fluorescent in situ hybridization

Fluorescent RNA in situ hybridization was performed based on the methods as previously described with some modifications [29]. To generate a sequence-specific probe of *Drosophila dpp*, the target region was amplified from the *dpp* transcript sequence with an accession number NM_057963.5 using the following primers: 5′-CACCAGAGTTGCAAGCGACCATG-3′ and 5′-CGCACTGTGTTGGCCGACTTG-3′ with a T7 promoter sequence (5′-TAATAC GACTCACTATAGGGAGA-3′) at the 5′ extremity. The Dig-labeled probe was synthesized using a T7 High Yield RNA Transcription Kit (Vazyme, Cat#TR101) and digoxigenin nucleotide mixes (DIG-11-UTP, Roche, Cat#11209256910) according to the manufacturer's instructions. For in situ hybridization, approximately 10 pairs of ovaries were dissected in RNAse-free 1× PBS, separating the ovarioles over the anterior 1/3 of each ovary, and immediately fixed in 4% formaldehyde in 1× PBST at 4°C overnight. The next day, the ovary samples were washed 5 min in 1× PBST 3 times and then treated with proteinase K (50 μg/ml in PBST) for 10 min at room temperature. After removing the proteinase K solution, samples were added to 2 mg/ml glycine solution to divert residual proteinase K activity, washed 3 times with 1× PBST, and refixed in 4% formaldehyde for 30 min, followed by incubation with prehybridization solution (50% formamide, 5× SCC, 0.1% Tween-20, 100 μg/ml salmon sperm DNA, and 100 μg/ml heparin) for 1 h at 56°C. Samples were then hybridized with the *dpp* probe overnight at 56°C with hybridization buffer. The next day, the ovaries were rinsed for 5 min in 1× PBST at 56°C 4 times, followed by incubation with anti-Dig-POD (1:200; Roche, Cat#11207733910) at room temperature for 30 min. The samples then developed in situ signals using a TSA Fluorescein system (AKOYA Biosciences, Cat#NEL701A001KT). Finally, the ovary samples were mounted in ProLong Gold Antifade Mountant with DAPI (Invitrogen, Cat#P36935), and the fluorescence images were captured using 63 ×/1.4 NA oil-immersion lenses with a Zeiss LSM 800 confocal microscope.

### Sample preparation and flow cytometry

*C587>RFP* and *C587>RFP*; *miR-124[6]* mutant flies were used to collect ECs, and *bab1>RFP* and *bab1>RFP*; *miR-124[6]* mutant flies were used to collect TF and Cap cells by using FACS to sort for cells expressing RFP. Specifically, ovaries from 3- to 5-day-old female adults were dissected in sterile 1× PBS. For each group, 100 to 120 pairs of ovaries in total were washed twice in calcium-free 1× PBS and incubated at room temperature for 20 min with intermittent gentle shaking in 700 μl of 0.5% trypsin and 0.25% collagenase in 1× PBS. Then, the dissociated ovaries were filtered through a 40 μm nylon mesh, and the individual cells were collected into 1.5 ml tubes. After centrifugation at 425 × *g* at 4°C for 5 min, the supernatant was removed, and the cell pellet was resuspended in 150 μl 1× PBS on ice for subsequent cell sorting.

FACS was performed with a MoFlo XDP high-speed cell sorter (ML99030; Beckman Coulter) using Summit 5.0 software (Beckman Coulter). The light source was a 561 nm diode laser (300 mW power), and the cytometer emission filter used was a 580/23 (FL6) to obtain the RFP

emission. Cells of each sample were sorted into 10 μl 1× PBS, snap frozen in liquid nitrogen, and then stored at −80˚C until use.

## Transcriptome analysis

Total RNA of sorted TF and Cap cells was extracted using an RNeasy Mini Kit (QIAGEN, Cat#74106), and purified mRNA was used to synthesize a cDNA library according to the protocol supplied by Illumina. Paired-end reads (150 bp) were obtained from the Illumina HiSeq X Ten platform. Raw sequence data were filtered by removing sequences with low-quality reads, adapter sequences and ambiguous bases and were used for alignment analysis with the *D. melanogaster* genomic sequences. The transcript expression level was calculated using the RSEM (v1.3.2) pipeline [98]. Differential expression analysis was performed by edgeR v3.30.3 on counts generated by RSEM [99]. Genes with an absolute value of log2 (expression fold change) of ≥1 in the comparison were identified as significantly up-regulated genes in the TF and Cap cells of the miR-124 mutant.

## Quantitative real-time PCR

To compare the relative RNA expression levels of *dpp* and *dally* in ECs between the $w^{1118}$ control and *miR-124* mutant flies, we sorted the cells via FACS as described above. Since we could not collect ECs using *C587>RFP* tools in the germaria with *EGFR* overexpressed in Cap cells with the help of *bab1-GAL4*, we alternatively dissected the most anterior part of the *Drosophila* ovarioles that covered the germarium and the early 3 egg chambers to detect the *dpp* levels.

Total RNA was isolated using the RNeasy Mini Kit (Qiagen, Cat#74106) and then reverse transcribed into cDNA using HiScript III RT SuperMix for qPCR (Vazyme, Cat#R223-01) according to the manufacturer's protocol. Quantitative real-time PCR (qRT-PCR) was performed in the QuantStudio3 Real-Time PCR System (Thermo Fisher Scientific) with the ChamQ SYBR qPCR Master Mix Kit (Vazyme, Cat#Q311-02). The primers used in this experiment are listed in S5 Table. Reactions were carried out for 30 s at 95˚C, followed by 40 cycles of three-step PCR for 10 s at 95˚C, 20 s at 55˚C, and 20 s at 72˚C. The RNA levels of the target genes were normalized to that of *actin 5C* mRNA, and their relative concentrations were determined using the $2^{-\Delta\Delta Ct}$ method [100].

## Plasmid construction and luciferase assays

The 130 putative target genes of miR-124 in *Drosophila* were predicted by TargetScanFly (http://www.targetscan.org/fly/) (S3 Table). In combination with the transcriptome data, the EGFR gene was selected to perform the luciferase assay for target validation due to the highest fold changes (Rank #1) of the up-regulated genes in the TF and Cap cells of the *miR-124[6]* mutant. The luciferase assay was carried out as described in our previous publication with some modifications [101]. Briefly, the 3′ UTR fragment (863 bp) of *EGFR* containing the miR-124 target site was PCR-amplified from *Drosophila* cDNA using specific primers and the 2 × Phanta Max Master Mix Kit (Vazyme, Cat#P525-02). Then, the fragments were cloned into the pMIR-REPORT miRNA Expression Reporter Vector (Ambion, Cat#AM5795) to generate a luciferase-*EGFR* reporter vector. Mutations at the miR-124 seed sequence binding region were introduced into the miRNA target site in the 3′ UTR of *EGFR* (GTGCCTT to CACGGAA) using a PCR-based method with specific primers. The mutant construct was used as the negative control. Both the normal (*EGFR*-3′ UTR$^{WT}$) and mutant constructs (*EGFR*-3′ UTR$^{MUT}$) were co-transfected with a synthesized miR-124 (5′-UAAGGCACGCG-GUGAAUGCCAAG-3′; Sangon Biotech) or a negative-control miRNA (miRCURY LNA microRNA Mimic Negative Control; Exiqon, Cat#479903–001) in HEK293 cells using

Lipofectamine 3000 (Invitrogen, Cat#L3000015), and the resulting luciferase activity of each combination was determined by a dual-luciferase assay at 48 h post-transfection to measure the interaction between miR-124 and *EGFR*. The Dual-Luciferase Reporter Assay Kit (Vazyme, Cat#DL101-01) was employed according to the manufacturer's instructions using the SpectraMax iD5 Multi Mode Microplate Reader (Molecular Devices). All primers used for vector construction are listed in S5 Table.

### Data analysis and statistics

Statistical analyses were performed in GraphPad Prism version 8.0 (GraphPad Software). Data were analyzed for statistical significance using unpaired two-tailed Student's *t* test (Fig 3B and 3C), one-way analysis of variance (ANOVA) along with Dunn's test (Figs 2H, 4I, 4J, 5D, 6E, S3, S6, S8, S10D, S15D, S15E, and S16B and S2 Table), ordinary one-way ANOVA along with Holm–Sidak's test (Fig 6I), two-way ANOVA with Sidak's multiple comparisons test (Figs 4D and S12B), two-way ANOVA with Dunnett's multiple comparisons test (S2 Fig), and Fisher's exact two-sided test (Figs 1D, 1G, 2C, 2D, 3G, 3K, 5C, 5H, 6H, S1C, S5C, S9, S14, and S16A). Error bars indicate the standard error of the mean (SEM), and the datasets are represented as the mean ± SEM. Significance values are indicated as follows: * $P < 0.05$, ** $P < 0.01$, and *** $P < 0.001$.

### Supporting information

**S1 Fig. miRNAs responsible for GSC maintenance. (A** and **B)** Germaria from wild-type **(A)** or a miRNA mutant **(B)** were stained with antibodies to Hts (red) and Vasa (green), together with DAPI (blue), to label spectrosomes or fusomes, germline cells, and nuclei, respectively. Scale bar: 10 μm. **(A)** The control germarium contained 2 GSCs (indicated by dashed circles). **(B)** Germarium of the *miR-971[KO]* mutant contained only 1 GSC (indicated by dashed circles). **(C)** Percentage of germaria with fewer than 2 GSCs in wild-type (*w^1118^*) and the designated homozygous miRNA mutants. The number of analyzed germaria is shown above each bar. Significance was determined by Fisher's exact two-sided test (** $P < 0.01$; *** $P < 0.001$). The raw data underlying panel C are available in S1 Data.
(TIFF)

**S2 Fig. Fertility is significantly reduced in *miR-124* mutants. (A)** The daily number of eggs laid by the *w^1118^* control (gray) and the *miR-124[6]* (red) or *miR-124[7]* (blue) mutants. **(B)** Hatching rate of the oviposited fly eggs in **(A)**. Five biological replicates were performed. Data represent the mean ± SEM. Significance was determined by two-way ANOVA with Dunnett's multiple comparisons test (* $P < 0.05$; ** $P < 0.01$; *** $P < 0.001$; ns: not significant). The raw data are available in S1 Data.
(TIFF)

**S3 Fig. Fluorescence intensity of dsRed expression in Fig 1H.** Quantification results on *miR-124* expression in TF and Cap cells (red circles), ECs (blue circles), and early FCs (green circles) per germarium. Each plot indicates the mean intensity from 5 cells for each germarium (*n* = 9 *Drosophila* germaria were examined for each group). Data are presented as the mean ± SEM. Significance was determined by Kruskal–Wallis one-way ANOVA with Dunn's test (** $P < 0.01$; *** $P < 0.001$). The raw data are available in S1 Data.
(TIFF)

**S4 Fig. *bab1-Gal4* is specifically expressed in TF and Cap cells, while *C587-Gal4* is specifically expressed in ECs. (A–A')** *UAS-mCD8RFP* and *bab1-GAL4* were used to indicate the *bab1-GAL4* expression pattern (red) in the germaria, which indicated *bab1-GAL4* is

specifically expressed in TF and Cap cells but not ECs. **(B–B')** *UAS-mCD8RFP* and *C587-GAL4* were used to indicate the *C587-GAL4* expression pattern (red) in the germaria, which indicated *C587-GAL4* is specifically expressed in ECs but not TF or Cap cells. **(A)** and **(B)** show the merging of the 2 channels of RFP and DAPI (blue); **(A')** and **(B')** show RFP-stained images in black and white. Scale bar: 10 μm.
(TIFF)

**S5 Fig. A temperature-sensitive conditional GAL4/GAL80ts system is used for gene expression specifically in adults. (A–B')** Animals carrying *UAS-mCD8RFP* and *GAL80*[ts]; *bab1-GAL4* were raised at 18˚C up to eclosion and then maintained at 18˚C **(A–A')** or 29˚C **(B–B')** for 5 days before ovary dissection. GAL4 activation of *UAS-mCD8RFP* was suppressed at 18˚C while was activated at 29˚C. **(A)** and **(B)** show the merging of the 2 channels of RFP and DAPI (blue); **(A')** and **(B')** show RFP stained images in black and white. Scale bar: 10 μm. **(C)** Animals carrying *UAS-miR-124* and *GAL80*[ts]; *bab1-GAL4* were raised at 29˚C up to eclosion and then maintained at 29˚C or 18˚C for the number of days indicated before ovary dissection. The percentage of germaria carrying 5 or more spectrosome-containing cells is shown, and the number of analyzed germaria is above each bar. Significance of 29˚C vs. 18˚C for the same time period was determined by Fisher's exact two-sided test (*** $P < 0.001$). The raw data underlying panel C are available in S1 Data.
(TIFF)

**S6 Fig. Fluorescence intensity of pMad staining in Fig 2E–2G.** Quantification results on pMad expression in GSCs (red circles) or extra cells (blue circles) per germarium of wild-type, *miR-124[6]*, and *miR-124[7]* mutants. In wild-type, there are no extra cells with pMad staining, so we choose the cells close to the GSCs for analysis. Each plot indicates the mean intensity from GSCs or extra cells with pMad staining for each germarium ($n = 30$ *Drosophila* germaria were examined for each group). Data are presented as the mean ± SEM. Significance was determined by Kruskal–Wallis one-way ANOVA with Dunn's test (** $P < 0.01$; *** $P < 0.001$; ns: not significant). The raw data are available in S1 Data.
(TIFF)

**S7 Fig. miR-124 is not required to maintain EC cellular processes and wrap germ cells. (A–C')** *UAS-mCD8RFP* and *C587-GAL4* were used to indicate EC cellular processes (red) in the germaria of wild-type, *miR-124[6]*, and *miR-124[7]* mutants. Similar to the control germarium **(A–A')**, differentiated germ cell cysts (labeled by Vasa, green) were normally wrapped by EC cellular processes in both *miR-124[6]* **(B–B')** and *miR-124[7]* **(C–C')** mutant germaria. **(A)**, **(B)**, and **(C)** show the merging of the 3 channels of RFP, Vasa, and DAPI (blue); **(A')**, **(B')**, and **(C')** show RFP stained images in black and white. **(D–D')** *UAS-mCD8RFP* driven by *C587-GAL4*-labeled EC membranes in *yki RNAi* germaria. EC cellular processes did not penetrate the interior in *yki* mutant germaria. **(D)** Shows the merging of the 2 channels of RFP and DAPI (blue); **(D')** shows RFP stained images in black and white. Scale bar: 10 μm.
(TIFF)

**S8 Fig. Fluorescence intensity of *dpp* in situ in Fig 3D–3F.** Quantification results on *dpp* signals in Cap cells (red circles) and ECs (blue circles) per germarium of wild-type, *miR-124[6]*, and *miR-124[7]* mutants. Each plot indicates the mean intensity from Cap cells or ECs for each germarium. Data are presented as the mean ± SEM. Significance was determined by Kruskal–Wallis one-way ANOVA with Dunn's test (*** $P < 0.001$; ns: not significant). The raw data are available in S1 Data.
(TIFF)

**S9 Fig. Limiting *dally* expression in ECs shows no effect on restoring germline differentiation defects in *miR-124* mutants.** Quantification results on the percentage of the germaria carrying 5 or more spectrosome-containing cells with different genotypes. The number of analyzed germaria is shown above each bar. Significance was determined by Fisher's exact two-sided test (*** $P < 0.001$; ns: not significant). Filled gray circles represent the presence, and empty brown circles represent the absence, of a given transgene. The raw data are available in S1 Data.
(TIFF)

**S10 Fig. Loss of miR-124 neither causes EC-to-Cap-cell transformation nor the expansion of endogenous Cap cell numbers. (A–C')** Germaria from wild-type females (**A–A'**, *w^1118^*), *miR-124[6]* (**B–B'**), and *miR-124[7]* (**C–C'**) mutants were immunostained for LamC (red) to identify Cap cells. Both in the control and *miR-124* mutant germaria, LamC is restrictedly expressed in the TF cells, Cap cells (yellow flat circle), and occasionally some muscle sheath cells (yellow arrow) but is not present in ECs. **(A)**, **(B)**, and **(C)** show the merging of the 3 channels of LamC, Vasa (green), and DAPI (blue); **(A')**, **(B')**, and **(C')** show LamC stained images in black and white. Scale bar: 10 μm. **(D)** Quantification results of the average numbers of Cap cells per germarium. Control germaria contain an average number of 5.27 Cap cells in 133 samples; the *miR-124[6]* and *miR-124[7]* mutant germaria contain an average number of 5.23 ($n = 137$) and 5.05 ($n = 105$) Cap cells, respectively. Data are presented as the mean ± SEM. Significance was analyzed by Kruskal–Wallis one-way ANOVA with Dunn's test (ns: not significant). The raw data underlying panel D are available in S1 Data.
(TIFF)

**S11 Fig. Loss of miR-124 does not change the expression pattern of *C587-GAL4* in ECs or *bab1-GAL4* in Cap and TF cells. (A–C')** *UAS-mCD8RFP* and *C587-GAL4* were used to indicate the *C587-GAL4* expression pattern (red) in the germaria of wild-type **(A)**, *miR-124[6]* **(B)**, or *miR-124[7]* **(C)** mutants. In both the control and *miR-124* mutant germaria, no ectopic expression of *C587-GAL4* was observed in Cap cells or TF cells (stained with LamC). **(D–F')** *UAS-mCD8RFP* and *bab1-GAL4* were used to indicate the *bab1-GAL4* expression pattern (red) in the anterior germaria of wild-type **(D)**, *miR-124[6]* **(E)**, or *miR-124[7]* **(F)** mutants, which were also immunostained for LamC (green) to identify Cap cells. Similar to that of the control germarium **(D–D')**, the expression pattern of *bab1-GAL4* was unchanged in both the *miR-124[6]* **(E–E')** and *miR-124[7]* **(F–F')** mutant germaria. **(A–F)** Show the merging of the 3 channels of RFP, LamC (green), and DAPI (blue); **(A'–F')** show RFP stained images in black and white. Scale bar: 10 μm.
(TIFF)

**S12 Fig. The confirmation of miR-124 target site in *Drosophila* S2 cells. (A)** The schematic diagram of recombinant plasmids generation and cell transfections. To generate the recombinant plasmid (pAc-*EGFR*^WT^), the sequence of *EGFR* that contained the coding region and the 3′ UTR fragment was amplified from cDNA of *D. melanogaster*, and then was cloned into the pAc5.1/V5-HisA insect expression vector (Invitrogen, Cat#V4110-20) at the KpnI site. The mutant construct (pAc-*EGFR*^MUT^) with mutation at the binding site of miR-124 was synthesized using Mut Express II Fast Mutagenesis Kit V2 (Vazyme, Cat#C214-02). *Drosophila* Schneider 2 (S2) cells were co-transfected with the plasmid expression vectors (pAc-*EGFR*^WT^ or pAc-*EGFR*^MUT^) and miR-124 or miRNA-NC at a 1:1 ratio using the Lipofectamine 3000 reagent (Invitrogen, Cat#L3000015) according to the manufacturer's instructions. All primers used for vector construction are listed in S5 Table. **(B)** The mRNA expression levels of *EGFR* were determined in S2 cells co-transfected with the plasmid expression vectors (pAc-*EGFR*^WT^ or pAc-*EGFR*^MUT^) and miR-124 or miRNA-NC using qPCR. All primers used for vector

construction are listed in S5 Table. Three biological replicates were performed. Data are presented as the mean ± SEM; significance was analyzed by two-way ANOVA with Sidak's multiple comparisons test (*** $P < 0.001$; ns: not significant). The raw data underlying panel B are available in S1 Data.
(TIFF)

**S13 Fig. The conservation of miR-124 target sites in different insect species.** The *EGFR* 3′ UTR in 20 insect species represent 6 orders, including Diptera (13), Lepidoptera (2), Hymenoptera (2), Coleoptera (1), Hemiptera (1), and Orthopterodea (1) were collected. The alignment results showed that the target sites of miR-124 in EGFR 3′ UTR are conserved in many *Drosophila* species, but have low or no similarity to other insects.
(TIFF)

**S14 Fig. A temperature-sensitive conditional GAL4/GAL80ts system was used for overexpressing the *EGFR* in TF and Cap cells.** Animals carrying *UAS-EGFR* and *GAL80ts; bab1-GAL4* were raised at 18˚C up to eclosion and then maintained at 18˚C or 29˚C for the number of days indicated before ovary dissection. The percentage of germaria carrying 5 or more spectrosome-containing cells is shown, and the number of analyzed germaria is above each bar. Significance of 18˚C vs. 29˚C for the same time period was determined by Fisher's exact two-sided test (*** $P < 0.001$). The raw data are available in S1 Data.
(TIFF)

**S15 Fig. Overexpression of EGFR in GSC niche cells induces ectopic pMad staining. (A–C')** Compared with the control germaria (**A–B'**, *UAS-EGFR* or *bab1-GAL4*) that contained pMad-labeled GSCs only (highlighted by dashed circles), more pMad (red)-positive cells (indicated by arrowheads) were present in *bab1-GAL4*-derived EGFR-overexpressing germaria **(C–C')**. **(A)**, **(B)**, and **(C)** show the merging of the 3 channels of pMad, Vasa, and DAPI (blue); **(A')**, **(B')**, and **(C')** show pMad stained images in black and white. Scale bar: 10 μm. **(D)** Quantification results of the average numbers of cells with pMad staining per germarium in different phenotypes. Germaria of $w^{1118}$ ($n = 90$), *UAS-EGFR* ($n = 96$), or *bab1-GAL4* ($n = 93$) contained average numbers of 1.96, 2.24, and 2.26 pMad cells, respectively. In contrast, the overexpression of *EGFR* in Cap cells caused ectopic pMad staining in the GSC progeny differentiation zone, with an average number of 3.88 in 114 samples. Data are presented as the mean ± SEM. Significance was analyzed by Kruskal–Wallis one-way ANOVA with Dunn's test (*** $P < 0.001$; ns: not significant). Filled gray circles represent the presence, and empty brown circles represent the absence, of a given transgene. **(E)** Quantification results on pMad expression in GSCs (red circles) or extra cells (blue circles) per germarium of wild-type, *UAS-EGFR*, *bab1-GAL4*, and *bab1>EGFR* mutants. Each plot indicates the mean intensity from GSCs or extra cells with pMad staining for each germarium ($n = 30$ *Drosophila* germaria were examined for each group). Data are presented as the mean ± SEM. Significance was determined by Kruskal–Wallis one-way ANOVA with Dunn's test (** $P < 0.01$; *** $P < 0.001$; ns: not significant). The raw data underlying panels D and E are available in S1 Data.
(TIFF)

**S16 Fig. Overexpression of EGFR in ECs causes no effect on GSC progeny differentiation. (A)** Percentage of the germaria carrying 5 or more spectrosome-containing cells with different genotypes. The number of analyzed germaria is shown above each bar. Significance was determined by Fisher's exact two-sided test (ns: not significant). **(B)** Quantification results of the average numbers of cells with pMad staining per germarium in different phenotypes. Germaria of $w^{1118}$ ($n = 92$), *UAS-EGFR* ($n = 85$), or *C587-GAL4* ($n = 88$) contained average numbers of 1.98, 2.18, and 2.19 pMad cells, respectively. Similarly, the overexpression of *EGFR* in

ECs caused no effect on ectopic pMad staining in the GSC progeny differentiation zone, with an average number of 2.01 in 89 samples. Data are presented as the mean ± SEM. Significance was analyzed by Kruskal–Wallis one-way ANOVA with Dunn's test (ns: not significant). Filled gray circles represent the presence, and empty brown circles represent the absence, of a given transgene. The raw data are available in S1 Data.
(TIFF)

**S17 Fig. The expression level of *Delta* (*Dl*) gene using the exact test of edgeR based on RNA-seq reads count in control vs. *miR-124[6]*.** The raw data are available in S1 Data.
(TIFF)

**S18 Fig. Characterization of miR-124 loss-of-function alleles used in this study.** The schematic drawing of miR-124 locus was modified from Sun and colleagues [48] and Weng and Cohen [49].
(TIFF)

**S1 Data. The underlying data related to the figures in this study.**
(XLSX)

**S1 Table. A collection of 90 miRNA mutant lines.**
(XLSX)

**S2 Table. Screening for GSC fate regulators in the homozygous viable miRNA mutants.**
(XLSX)

**S3 Table. Transcriptome analysis of the putative target genes of miR-124.**
(XLSX)

**S4 Table. Average number of ECs with ectopic dpp expression in wild-type and *miR-124* mutants.**
(XLSX)

**S5 Table. Primers and nucleotide information in this study.**
(XLSX)

## Acknowledgments

We thank Dr. Daniel Kalderon for critical comments and edits to improve the manuscript, Drs. Shuai Zhan and Gangqi Fang for analysis of the transcriptome data. We also thank Drs. Eric Lai, Zheng Guo, Zhaohui Wang, Zongzhao Zhai, and the Bloomington Stock Center for providing genetic reagents; the Developmental Studies Hybridoma Bank (DSHB) for providing antibodies; and the FACS platform in College of Life Sciences, Zhejiang University, for isolating specific cell populations.

## Author Contributions

**Conceptualization:** Jianhua Huang.

**Data curation:** Jiani Chen, Chaosqun Li, Yifeng Sheng, Yueqi Lu, Zhiguo Liu.

**Formal analysis:** Jiani Chen, Yifeng Sheng, Junwei Zhang, Zhi Dong.

**Funding acquisition:** Jiani Chen, Xuexin Chen, Jianhua Huang.

**Investigation:** Jiani Chen, Chaosqun Li, Yifeng Sheng, Junwei Zhang, Zhiwei Wu, Jianhua Huang.

**Methodology:** Jiani Chen, Junwei Zhang, Yueqi Lu, Zhiguo Liu.

**Resources:** Jiani Chen, Junwei Zhang.

**Software:** Jiani Chen, Yifeng Sheng, Junwei Zhang, Zhi Dong, Qichao Zhang.

**Supervision:** Jianhua Huang.

**Validation:** Jiani Chen, Chaosqun Li, Junwei Zhang, Lan Pang, Zhiwei Wu, Yueqi Lu, Zhiguo Liu, Qichao Zhang, Xueying Guan.

**Visualization:** Jiani Chen, Qichao Zhang, Xueying Guan, Xuexin Chen.

**Writing – original draft:** Jiani Chen, Jianhua Huang.

**Writing – review & editing:** Jiani Chen, Jianhua Huang.

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
