## [Editor Report · Decision Letter 0]

13 Jul 2023

Dear Dr Huang, 

Thank you for submitting your manuscript entitled "EGFR- and microRNA-mediated communication between adjacent niches promotes stem cell self-renewal and differentiation" for consideration as a Research Article by PLOS Biology.

Your manuscript has now been evaluated by the PLOS Biology editorial staff as well as by an academic editor with relevant expertise and I am writing to let you know that we would like to send your submission out for external peer review. As a note, while we are interested in the study, we have yet to make a firm call about whether the study represents a sufficient advance for PLOS Biology - and so we will be looking for strong reviewer support about the fit of this study for PLOS Biology.

Before we can send your manuscript to reviewers, we need you to complete your submission by providing the metadata that is required for full assessment. To this end, please login to Editorial Manager where you will find the paper in the 'Submissions Needing Revisions' folder on your homepage. Please click 'Revise Submission' from the Action Links and complete all additional questions in the submission questionnaire.

Once your full submission is complete, your paper will undergo a series of checks in preparation for peer review. After your manuscript has passed the checks it will be sent out for review. To provide the metadata for your submission, please Login to Editorial Manager (https://www.editorialmanager.com/pbiology) within two working days, i.e. by Jul 17 2023 11:59PM.

Kind regards,

Luke

Lucas Smith, Ph.D.

Senior Editor

PLOS Biology

lsmith@plos.org

---

## [Decision Letter · Decision Letter 1]

1 Sep 2023

Dear Dr Huang,

Thank you for your patience while your manuscript "EGFR- and microRNA-mediated communication between adjacent niches promotes stem cell self-renewal and differentiation" was peer-reviewed at PLOS Biology. It has now been evaluated by the PLOS Biology editors, an Academic Editor with relevant expertise, and by several independent reviewers. 

In light of the reviews, which you will find at the end of this email, we would like to invite you to revise the work to thoroughly address the reviewers' reports.

As you will see below, the reviewers find the work generally well done and interesting. However, they indicate that some of the mechanistic conclusions of the study are not yet fully supported by the data. We think that the reviewer concerns will need to be thoroughly addressed before we can consider your manuscript can be considered further for publication at PLOS Biology. As a last note, Reviewer 1 notes that the bab1-Gal4 line is not specific for TF and cap cells only - noting that bab1 expression has been observed in escort cells. We have discussed this point with the other reviewers and with the Academic Editor, and on balance, we agree with Reviewer 1 that additional experimental work would be needed to address this point and to strength the study. For example key experiments should be repeated with another TF and cap cell specific line, to validate these conclusions.

Given the extent of revision needed, we cannot make a decision about publication until we have seen the revised manuscript and your response to the reviewers' comments. Your revised manuscript is likely to be sent for further evaluation by all or a subset of the reviewers.

**IMPORTANT - SUBMITTING YOUR REVISION**

*Re-submission Checklist*

*Published Peer Review*

*PLOS Data Policy*

*Blot and Gel Data Policy*

Sincerely,

Lucas

Lucas Smith, Ph.D.

Senior Editor

PLOS Biology

lsmith@plos.org

REVIEWS:

Reviewer #1: In this work, the authors study the transition from an undifferentiated stem cell to a differentiating progeny using the Drosophila ovarian niche as their model system. They report that miR-124 acts to repress EGFR transcription in some somatic cells of the niche (terminal filament cells and cap cells). Increased EGFR signaling in miR-124 mutant niche cells induced Delta (one of the Notch ligands) transcription. Reception of the Dl signal by Notch in the escort cells activates/enhances dpp transcription, thus preventing germline differentiation.

I believe the biological question under research to be of wide interest. The MS is well written and the results are carefully documented. I do however have some concerns regarding the conclusions reached by the AUs.

1- The use of the bab1-Gal4 line to address TF + cap cell versus escort cell realm of action. The data presented by the authors seem to indicate that the bab1-Gal4 line used in this study is expressed in TF and cap cells only. The authors cite a number of previous publications that reinforce their findings. There are in the literature however contradictory results regarding this line (which the authors do not even acknowledge), with some authors unequivocally showing that some bab1-Gal4 lines are expressed clearly in escort cells. In fact, some of the publications used by the authors to confirm the bab1-Gal4 expression in TF and cap cells only do show/mention that this line is also expressed in escort cells (see, for instance, Bolivar et al 2006; Hamada-Kawaguchi 2014). Since this is a key point in this work (is miR-124 acting in the escort cells in addition to TF and cap cells?) I believe the authors should check with a TF and cap cell specific line the validity of their results. This is particularly important considering (1) that miR-124 is expressed in the most anterior escort cells too, (2) the data in Fig. 2D and (3) my concern number 2.

2- According to their model, cap cells express miR-124 to repress EGFR transcription (and thus excess activation of the pathway). In fact, EGFR mRNA has a target for miR-124 and knocking down EGFR transcription using bab1-Gal4 in a miR-124 mutant background rescues the extra-spectrosomes phenotype. Increased EGFR activity induces extra Delta in miR-124 mutant cap cells, which in turn signals to adjacent escort cells via the Notch receptor pathway. Notch activation in escort cells (in theory, the ones adjacent to the cap cells) would then upregulate dpp transcription. Allegedly, this increase in dpp would prevent CB differentiation. My problem with this model is that the anterior escort cells are known to be of different quality to more posterior ones, and they express dpp. Thus, are the authors implying in their model that miR-124 mutants express more dpp only in these anterior escort cells? But they show in figure 4 that more posterior miR-124 mutant escort cells also express dpp. If their model is correct, how does Dl in the cap cells reach to posterior escort cells? I may add that, if miR-124 acts not only in cap cells but also in anterior escort cells (as suggested by their own data, Fig 2D), this could explain why dpp is expressed in more posterior escort cells in the mutant condition. In summary, I am not convinced that the authors have fully proven that miR-124 acts only in TF and cap cells and that the increase in Dl levels in mutant cap cells is responsible for the observed phenotypes.

3- Figs 1, 2 and/or 3 could be combined in just one or two figures.

4- Fig. 5: Did the authors find other target genes (apart from the 130 known targets) that were not predicted to be bound by miR-124 (may be indirect targets)?

5- Line 66: consider prospective "daughter CB".

6- Lines 127, 128: The spectrosome in GSCs is not always spherical. It follows a well-described cycle along the cell cycle.

7- Fig. 1H': Consider a black and white version of this panel to enhance the contrast.

8- Ref. 57: These authors showed bab1-Gal4 expression in escort cells.

9- Line 193: Data in Fig 2D indicates that miR-124 does have a function in escort cells. I do not understand why the authors neglect this finding in their conclusions.

10- Line 200: Why do the authors use reference 60 to cite pMad expression in GSCs? It was reported previously to this 2020 paper. Consider Kai + Spradling 2004 or Zhao et al Ageing Cell 2008.

11- Legend to Fig S6: substitute mutant for mutation.

12- To look at escort cell projections I am not sure that the c587-Gal4 line is the best option. I would suggest the authors to generate MARCM clones with CD8:GFP to look at the shape of individual or small clusters of escort cells in control and mutant conditions.

13- Lines 229 and 230: This is slightly confusing. First, as indicated above, it is not clear to me that the focus of the action of miR-124 is in TF and Cap cells. Second, if the authors are referring to an alternative explanation for their 124 results, I would recommend them to rephrase the sentence.

14- Lines 264 and 265: Again, I would be more cautious about the specificity issue.

15- Fig 5E', F' and G': Consider black and with panels to enhance contrast

16- Lines 334-342: Please be careful. Since bab1-Gal4 is expressed (at least) in TF and cap cells, you cannot conclude that "miR-124 normally acts … in cap cells". The same comment applies to line 346.

17- Lines 348-349: Revise grammar "was significantly increased its expression".

18- Line 358: Typo in Fig 7E (germarium, not gerarium).

Reviewer #2: General comments

Chen et al first conducted the screening of 90 miR mutants, and among those, they identified miR-124 as an essential factor required for proper differentiation of GSCs. miR-124 is known to function for neuroblast proliferation in Drosophila as well as for tissue stem cells in other animals. Loss of miR-124 activity in Cap cells, the GSC niche, led to differentiation defect of germ-line cells, caused by upregulation of Dpp activity through notch signaling in Escort cells, the another due to unchecked epidermal growth factor receptor (EGFR) in Cap cells. This activates anti-differentiation signals in neighboring cells through Notch signaling, refining the boundary between self-renewal and differentiation niches. This manuscript describes a novel mechanism how the GSC niche regulates Dpp signaling to promote GSC differentiation. Of note, the authors found that Cap cells, the GSC niche, communicate with the neighboring escort cells (ECs), which promote differentiation of CBs and cysts (differentiation niche): the expression in DI is repressed in Cap cells by mir-124, inhibiting notch signaling activation to avoid Dpp expression ECs. The manuscript is well-written, and the logical flow is understandable. In Discussion, a diverse array of related aspects, though not the primary focus of this study, were adequately discussed. Though some of the proper controls in several genetical analyses should be conducted, this study would be a general interest to be published in PLOS Biology.　 

Major points

1. The schematic drawing of miR-124 locus, the main focus of this study, should be shown somewhere in the figure, including information of loss-of function alleles examined in this study.

2. Figure 1 and S2 clearly showed the defects in GSC maintenance and/or proliferation of the examined miR mutants. However, this study did not describe or argued about fertility. Egg numbers and the hatching rate should be described for the main focus, miR-124 mutant. And if it's not affected, it should be argued in Discussion why those defects in early differentiation does not affect fertility. 

3. Page 10, Lane 284, MiRNAs usually cause mRNA deadenylation and decay [67]. miRNAs could perturb efficient translation, resulting in failure of the protein expression of the target. This statement is too simplified and needs to be rephrased.

4. The authors nicely showed that luciferase reporter containing 3'UTR of fly EGFR in HEK293 was repressed in the presence of miR-124, and the repression was relieved upon substitution of the nucleotide sequence of the potential miR-124 target site. However, it does not exclusively indicate that the direct target of miR-124 is EGFR 3'UTR in fly vivo. Hence, the subheading, EGFR is a target of miR-124 in Cap cells, is currently overstatement. Preferably it needs to be examined in vivo by making transgene containing reporter and EGFR 3'UTR and its mutant in miR-124 mutant background. In addition, the rescue experiment for pERK should be conducted; the expression of miR-124 in Cap and TF, but not in EC, in miR-124 mutant background should reduce the pERK signal in EC.

5. Related with the experiment using HEK293 cells above, I wonder if overexpression of miR-124 in Drosophila S2 cells can repress endogenous or transfected EGFR. If this works, mutation analysis could be done in S2 cells. Because there is no report for the direct regulation of EGFR by miR-124, this is an important piece of the data in this study. In addition, is the target site of miR-124 in EGFR 3'UTR conserved? If so, it would be better to show the alignment among different species. 

6. What happens when EGFR is over-expressed in ECs? 

Minor points

1. Page 6, Line 131-132, suggesting a role in the stem cell niche.: the observation alone does not suggest a role in the niche, but suggest a role for GSC maintenance and/or proliferation.

2. Fig1B, D, Fig 3A and many others: the labels should be w1118, not W1118.

3. For a clearer presentation, the staining data should be presented in black and white when only one staining is shown; for example, Figure 1H', 3A'-C', 5E'-G', 7A'-D' and elsewhere applicable in Supplementary figures.

4. Page 11, Lane 325-326, EGFR overexpression also significantly increased dpp RNA levels in extracts from the germarium and early egg chambers (Fig 6D).: dpp level in RNA isolated from the germarium and early egg chambers.

5. Related to the Major Point #4, Page 11, Line 340-342, These results indicate that miR-124 normally acts directly on a defined element of the EGFR 3' UTR to limit EGFR signaling in Cap cells and thereby prevent ectopic Dpp expression in ECs and delayed germline differentiation.: This is overstatement, because the authors did not show the perturbation of the EGFR repression by mutation of miR-124 target site in 3'UTR of EGFR in vivo. A possibility of indirect regulation of EGFR by miR-124 cannot be excluded.

Reviewer #3: In this study, Chen and colleagues identified one microRNA (miR124) expressed in the somatic niche of germline stem cells in Drosophila. They show that miR124 is required for the differentiation of the posterior daughter cell after the stem cell division. They further demonstrate that miR124 acts through a signaling relay mechanism, which starts by inhibiting EGFR in the anterior niche cells. The inhibition of EGFR then inhibits the synthesis of Delta, downregulating Notch activity in the posterior niche cell, which then downregulates Dpp. Ultimately downregulation of Dpp allows for the differentiation of the posterior daughter cell. 

The quality of the data is of very high standards, whether they are microscopy, genetics or transcriptomic analysis. The results are convincing, well-controlled, and interpreted rigorously. It is an impressive feat given that the authors use a wide variety of techniques to decipher the underlying mechanisms. I don't have any major criticisms regarding the data and methods used as presented.

Overall, the authors' conclusions and model fit well with the current paradigm of how the GSC niche works in Drosophila. There is one anterior somatic niche maintaining undifferentiated GSCs and one posterior niche "allowing" differentiation rather than "inducing" differentiation. The EGF and Notch pathways were already identified as important for niche establishment and functions. However, the scope of this study could be more important if they could integrate their results with previously published data. In particular, in (Ward et al, Current Bio 2006) or (Song et al. Development, 2005), Delta has been proposed to signal from germline cells to somatic niche cells to regulate the formation and function of the niche. In my opinion, it is difficult to see how delta signaling from germline cells could fit in the authors' model on Figure 7J. It would be very important for the significance of this study to, at least, discuss it. 

Other comments:

1) In figure 2, the authors use an elegant combination of bab1-Gal4 and Gal80ts, but not in figure 6, when KD or OE of EGFR. The same is true on Figure 7. Is there a specific reason?

2) miR124 mutant are viable, but are they fertile? Is there any other defect during oogenesis?

---

## [Decision Letter · Decision Letter 2]

12 Jan 2024

Dear Dr Huang,

Thank you for your patience while we considered your revised manuscript "EGFR- and microRNA-mediated communication between adjacent niches promotes stem cell self-renewal and differentiation" for publication as a Research Article at PLOS Biology. This revised version of your manuscript has been evaluated by the PLOS Biology editors, the Academic Editor and the original reviewers.

Based on the reviews, we are likely to accept this manuscript for publication, as suggested by the 3 reviewers. We do suggest that you consider Reviewer 3's last point and update the discussion and your model as needed. Please also make sure to address the following data and other policy-related requests, as these will be required before we can accept your study.

**IMPORTANT: Please address the following editorial requests: 

1) TITLE: After some discussion within the team, we would like to suggest the title be modified to include a bit more of the specifics fo the study. If you agree (and if supported), we suggest it be changed to something like: 

"Communication between the stem cell niche and an adjacent differentiation niche through miRNA and EGFR signalling orchestrates exit from the stem cell state in the Drosophila ovary"

2) FINANCIAL DISCLOSURES: Please update your financial disclosures statement, in our electronic system, to describe the role of any sponsors or funders in the study design, data collection and analysis, decision to publish, or preparation of the manuscript. If the funders had no role in any of the above, include this sentence at the end of your statement: "The funders had no role in study design, data collection and analysis, decision to publish, or preparation of the manuscript."

3) DATA: You may be aware of the PLOS Data Policy, which requires that all data be made available without restriction: http://journals.plos.org/plosbiology/s/data-availability. For more information, please also see this editorial: http://dx.doi.org/10.1371/journal.pbio.1001797

We see that you have provided your transcriptomic data on a repository, and that some relevant underlying data is contained within table S6. However, to be compliant with our policy we need you to provide all of the underlying data related to the figures in your study. Note that we do not require all raw data. Rather, we ask that all individual quantitative observations that underlie the data summarized in the figures and results of your paper be made available in one of the following forms:

a. Supplementary files (e.g., excel). Please ensure that all data files are uploaded as 'Supporting Information' and are invariably referred to (in the manuscript, figure legends, and the Description field when uploading your files) using the following format verbatim: S1 Data, S2 Data, etc. Multiple panels of a single or even several figures can be included as multiple sheets in one excel file that is saved using exactly the following convention: S1_Data.xlsx (using an underscore).

b., Deposition in a publicly available repository. Please also provide the accession code or a reviewer link so that we may view your data before publication. 

>>Regardless of the method selected, please ensure that you provide the individual numerical values that underlie the summary data displayed in the following figure panels as they are essential for readers to assess your analysis and to reproduce it:

Fig 2H; Fig 3B-C; Fig 4D,I-J; Fig 5; Fig 6E,I; Fig S2; Fig S3; Fig S6; Fig S8; Fig S12b; Fig 15D-E; Fig 16B; Fig 17;

>>Please also ensure that figure legends in your manuscript include information on where the underlying data can be found, and ensure your supplemental data file/s has a legend.

>>Please ensure that your Data Statement in the submission system accurately describes where your data can be found.

4) CODE: Per journal policy, if any code was generated that is important to support the conclusions of your manuscript, we would require that you make it available without restrictions upon publication. Please ensure that any code is sufficiently well documented and reusable, and that your Data Statement in the Editorial Manager submission system accurately describes where your code can be found.

We expect to receive your revised manuscript within two weeks. 

*Published Peer Review History*

*Press*

Sincerely,

Lucas

Lucas Smith, Ph.D.

Senior Editor,

lsmith@plos.org,

PLOS Biology

Reviewer remarks:

Reviewer #1: I appreciate the care and thoughtfulness of the AUs in their reply to my comments. I am happy with the current version of the MS, which has came out as a round and very interesting story. Congratulations.

Reviewer #2: The authors have satisfactorily addressed most of my concerns, though the in vivo analysis for EGFR 3'UTR has not been addressed. However, the authors examined that point in S2 cells and investigated the conservation of 3'UTR. In addition, other genetical analyses strongly support their claim. I personally think the revised manuscript is, in principal, now ready for publication.

Reviewer #3: In this revised manuscript, the authors have addressed my main concerns experimentally and successfully. However, in contrast to the authors, I am still puzzled by how Escort Cells can integrate Delta signals from Cap cells AND Germline cells at the same time? It is not shown in their final model (Figure 7J). It is nonetheless a minor point in this impressive piece of work.

---

## [Editor Report · Decision Letter 3]

22 Jan 2024

Dear Dr Huang,

Thank you for the submission of your revised Research Article "Communication between the stem cell niche and an adjacent differentiation niche through miRNA and EGFR signalling orchestrates exit from the stem cell state in the Drosophila ovary" for publication in PLOS Biology and thank you for addressing the last reviewer and editorial comments in this revision. On behalf of my colleagues and the Academic Editor, Yukiko M Yamashita, I am pleased to say that we can in principle accept your manuscript for publication, provided you address any remaining formatting and reporting issues. These will be detailed in an email you should receive within 2-3 business days from our colleagues in the journal operations team; no action is required from you until then. Please note that we will not be able to formally accept your manuscript and schedule it for publication until you have completed any requested changes.

PRESS

Sincerely, 

Lucas Smith, Ph.D.

Senior Editor

PLOS Biology

lsmith@plos.org